

**Rising atmospheric CO₂ concentrations: the overlooked factor promoting SW Iberian**
**Forest development across the LGM and the last deglaciation?**
Gomes, Sandra.D.[a,b,c*]
Fletcher, William.J.[a]
Stone, Abi[a]
Rodrigues, Teresa[b,c]
Rebotim, Andreia[b,c]
Oliveira, Dulce [b,c]
Sánchez Goñi, Maria. F.[d,e]
Abrantes, Fatima[b,c]
Naughton, Filipa[b,c]
[a]Quaternary Environments and Geoarchaeology, Department of Geography, School of
Environment, Education and Development, The University of Manchester, Manchester,
Oxford Road, Manchester, M13 9PL, United Kingdom;
[b]Divisão de Geologia e Georecursos Marinhos, Instituto Português do Mar e da Atmosfera
(IPMA), Rua Alfredo Magalhães Ramalho 6, 1495-006 Lisboa, Portugal;
[c]Centro de Ciências do Mar (CCMAR), Universidade do Algarve, Campus de Gambelas, 8005
- 139 Faro, Portugal;
[d] École Pratique des Hautes Études, EPHE, PSL Université, Paris, France;
[e]Environnements et Paléoenvironnements Océaniques et Continentaux, UMR 5805,
Université de Bordeaux, Pessac, France.
*Corresponding author: E-mail: sandra.domingues@manchester.ac.uk (Sandra Domingues
Gomes); Address: Quaternary Environments and Geoarchaeology, Department of
Geography, School of Environment, Education and Development, The University of
Manchester, Manchester, Oxford Road, Manchester, M13 9PL, United Kingdom





**Abstract:**


Across the last deglaciation, the global atmospheric concentration of carbon dioxide ($pCO_2$)
increased from ~180 to ~280 ppm. However, the impact of $pCO_2$ changes on vegetation
across the last deglaciation remains poorly understood. Under full glacial low $pCO_2$, plants
experienced constraints on photosynthesis. Therefore, a significant reduction in $pCO_2$
limitation should have impacted local and regional vegetation dynamics across deglaciation.
We hypothesise that deglacial $pCO_2$ rise could have (1) led to a gradual reduction of the
physiological constraint promoting forest response when moisture availability was sufficient,
despite low temperatures; and (2) amplified the response of forest development to warmer
and wetter conditions. The high-resolution analysis of terrestrial (pollen, $C_{29}:C_{31}$ organic
biomarker) and marine (alkenone-derived Sea Surface Temperature, $C_{37:4}$%, and long-chain
n-alkanes ratios) indicators, using a direct land-sea direct comparison, in the Iberian margin
site Integrated Ocean Drilling Program (IODP) U1385 ("Shackleton site") throughout the Last
Glacial Maximum (LGM) and last deglaciation allowed us to track and compare the changes
with shifts in global $pCO_2$. The LGM is characterised by a grassland-heathland mosaic type
ecosystem, triggered by cool and moderately humid conditions but low $pCO_2$ levels may have
exacerbated physiological drought and suppressed forest development. During Heinrich
Stadial 1 (HS1)the coldest and most arid conditions combined with sustained low $pCO_2$ values
precluded forest development and resulted in the dominance of Mediterranean steppe or semi-
desert vegetation. The Bølling-Allerød (BA) is characterised by a temperature optimum
(warmest SSTs and forest development) and variable moisture condition, while increasing
$pCO_2$ which contributed to the highest levels of forest development. Within the BA, significant
SW Iberian forest development occurred at ~15 cal kyr B.P. associated with an increase in
$pCO_2$ above 225 ppm. During the Younger Dryas (YD), cool temperatures combined with
sufficient moisture availability allowed the maintenance of a grassland-forest mosaic, the
increasing values of $pCO_2$ in this period should have offset the low temperature as well as the
moisture levels and allow the forest to persist. The overlooked role of $pCO_2$ could explain an
intriguing feature observed in Site U1385 and other Iberian margin records, namely the near
absence of forest during the LGM and HS1 but relatively high forest cover during the YD. Our
study aims to shed light on the influence of climatic factors (temperature and moisture
availability) together with $pCO_2$ as thresholds on forest response to deglacial climate changes
across the Iberian Peninsula.

Keywords:
Iberian margin; Deglaciation; LGM; Direct sea-land comparison; climatic space; Climatic
parameters vs $pCO_2$; Forest development; Pollen analysis
**1. Introduction**
The last deglaciation is characterised by a series of "classic" abrupt climate events, being one
of the periods widely studied for its particular succession of cold and warm events (Alley and
Clark, 1999; Lynch-Stieglitz, 2007; Fletcher et al., 2010a; 2010b; Salgueiro et al., 2014;
Marcott et al., 2014; Martrat et al., 2014; Naughton et al., 2016; Ausín et al., 2020). While
some records based on direct sea-land comparison are available for SW Iberian margin
(Boessenkool et al., 2001; Turon et al., 2003; Chabaud et al., 2014; Oliveira et al., 2018;
Naughton et al., 2019), few exist that cover the entire deglaciation, and none have the required
resolution or chronological precision to detect abrupt vegetation and climate shifts in detail.
The high temporal resolution, direct sea-land comparison provided by Site U1385 enables,



therefore, the detection of significant vegetation and climatic shifts in SW Iberia across the last
deglaciation.
The interactions between the lithosphere, hydrosphere (oceanic and terrestrial), cryosphere
and atmosphere during extreme climate events are crucial to understanding the climate
system behaviour. The last deglaciation, from 21 to 6 ka, was accompanied by a global
temperature increase of 5 to 10˚C, depending upon the latitude (Bard et al., 1987; Alley and
Clark, 1999; Clark et al., 2012), although the warming was not continuous. Two major abrupt
climatic cooling episodes, associated with ocean-atmospheric perturbations were
superimposed on the warming trend, the Heinrich Stadial 1 (HS1) and the Younger Dryas
(YD), bracketing the intervening Bølling-Allerød (BA) interstadial. During the last deglaciation,
global atmospheric concentrations of carbon dioxide ($pCO_2$) increased from ~180 ppmv to 280
ppmv (Monnin et al., 2001; Shakun et al., 2012;  Marcott et al., 2014), being among the highest
amplitude shifts in $pCO_2$  of the last 800,000 years  (Lüthi et al., 2008). The high-temporal
resolution West Antarctic Ice Sheet Divide ice core furthermore shows three main rapid (< 200
years) $pCO_2$ rises, each of ~10 to 15 ppmv, which took place at the end of HS1; within the BA
and at the onset of the YD (Marcott et al., 2014).
The role of $CO_2$ as a climate driver throughout the ice ages is still intensely debated; however,
it has been mainly considered as either (1) a primary driver of the climatic changes, in which
the $CO_2$ led the temperature records in Northern Hemisphere ( Shakun et al., 2012; Marcott
et al., 2014); (2) an amplifier responding as positive feedback to the warming (Alley and Clark,
1999; Clark et al., 2012); or (3) as a consequence rather than a cause of climatic changes
(Denton et al., 2010). Besides its impact on climate, the physiological influence of $pCO_2$ as a
limiting factor over plant development on Quaternary timescales has also been discussed
(Cowling and Sykes 1999; Crucifix et al., 2005; Ward et al., 2005; Gerhart and Ward, 2010;
Harrison and Sanchez Goñi, 2010). However, its role is often neglected, for example in
vegetation-based climate reconstructions which do not account for $pCO_2$ effects (e.g. Elenga
et al.; 2000; Sánchez Goñi et al., 2002; Peyron et al., 1998; Fletcher et al., 2010a; Bartlein et
al., 2011; Tarroso et al., 2016).
The annual exchange of $CO_2$ between the atmosphere and biosphere due to photosynthetic
activity corresponds to more than one-third of the total $pCO_2$ stored in the atmosphere
(Farquhar and Lloyd, 1993). The study of increased plant growth and global vegetation
greening under higher concentrations of $pCO_2$ ($CO_2$ fertilisation) is very topical within
discussions of current global climate change (e.g. Piao et al., 2019) whilst the inverse scenario
(low $pCO_2$) has received less attention. The influence of lowering $pCO_2$ on vegetation has
been examined in coupled climate-vegetation models (e.g. Harrison and Prentice, 2003; Sitch
et al., 2003; Crucifix et al., 2005; Ramstein et al., 2007; Wu et al., 2007; Prentice and Harrison
2009; Bartlein et al., 2011; Woillez et al., 2011; Svenning et al., 2008; 2011; Claussen et al.,
2013; Prentice et al., 2017; Shao et al., 2018; Cao et al., 2019). It has been suggested that
$pCO_2$ changes play an essential role on the development of vegetation (Wu et al., 2007), its
coverage (e.g. Harrison and Prentice 2003; Woillez et al., 2011; Harrison et al., 2016; Cao et
al., 2019),  vegetation productivity (Claussen et al., 2013) and water use efficiency (WUE)
(Polley et al., 1995; Cramer et al., 2001).
The role of $pCO_2$ in plant physiology is well known, in particular during photosynthesis, but the
magnitude of its influence on the composition and distribution of past vegetation remains
poorly understood. Under low $pCO_2$ concentrations, stomatal conductance and stomatal
density must increase to maintain an adequate $CO_2$ gradient between the atmosphere and the
leaf during photosynthesis. The evaporative demand increases and higher amounts of water
are lost through transpiration, reducing WUE and imposing a physiological drought (Street-





Perrot et al., 1997; Körner, 2000). One should expect that the $CO_2$ limitations on plant
development are, and were, not globally or temporally uniform, depending mainly on the
regional level of water-stress. The particular influence of $CO_2$ limitations in arid and semi-arid
areas is highlighted by evidence for global greening of arid areas due to $CO_2$ fertilisation
(Randall et al., 2013). Global evidence supports an atmospheric $CO_2$ fertilisation effect,
especially trees growing in drought-stressed conditions which benefit from increased WUE to
enhance growth (Huang et al., 2007). Nevertheless, at local scales, nutrient limitations may
limit the response of vegetation to rising $CO_2$ (e.g. Tognetti et al., 2008). The Mediterranean
region, with its characteristic annual hydrological deficit and seasonal water stress, is a key
place for exploring the potential role of $pCO_2$ limitation on vegetation growth. Therefore, past
vegetation dynamics in this region may be considered as a significant (inverse) analogue to
understand the current impact of increasing temperature and $pCO_2$ within semi-arid and arid
ecosystems.
Vegetation changes across the Iberian Peninsula for the last deglaciation as recorded in
palaeoecological proxies are traditionally interpreted as a result of the combined effects of
temperature, precipitation and evaporation changes (e.g. Peyron et al., 1998; Carrión et al.
2002; Sánchez Goñi et al., 2002; Combourieu Nebout et al., 2009; Dormoy et al., 2009;
Fletcher et al., 2010a; Arranbari et al., 2014; Bartlein et al., 2011; Naughton et al., 2011; 2019;
Tarroso et al., 2016).   The majority of climate reconstructions and simulations for glacial
periods based on vegetation records do not consider the influence of $CO_2$ and may, therefore,
be biased when the effect of $pCO_2$ is not included. Palaeovegetation (pollen-based) data is
vital for testing climate model simulations, both as temporal trends and reconstructions of
spatial distributions (Prentice et al., 1992; 2001; Jolly and Haxeltine, 1997; Harrison and
Prentice, 2003; Bartlein et al., 2011; Prentice et al., 2011; Harrison et al., 2016; Cao et al.,
2019). Within this, reconstructions of vegetation assemblage (Elenga et al., 2000; Svenning
et al., 2008) and quantitative estimates of climatic variables (Wu et al., 2007; Svenning et al.,
2008; Prentice et al., 2017) are both critical. It is essential to recognise that $pCO_2$ is correlated
with WUE or the balance between carbon assimilation and transpiration (water loss).
Neglecting this influence may contribute to the unreliability of precipitation reconstructions,
specifically underestimation of past precipitation under full glacial conditions  (Jolly and
Haxeltine, 1997; Cowling and Skyes, 1999; Gerhart and Ward, 2010; Prentice et al., 2017;
Cleator et al., 2020). In contrast, the influence of $CO_2$ on forest development in the southwest
Iberian Peninsula under interglacial conditions may be neglible compared with precipitation
changes, as recently revealed by transient model experiments (Oliveira et al., 2018).
The need for additional regional-based palaeoecological studies, such as for the southwest
Iberian Peninsula, is highlighted in a recent model-data comparison using the BIOME4 model
and a biome-scale reconstruction compiled from pollen records across the Northern
Hemisphere (> 30°N), which reveals a level of unexplained variability in patterns across both
space and time (Cao et al., 2019). Detailed pollen assemblage datasets may provide key
insights into other factors than temperature, precipitation and potential evaporation that drive
changes in vegetation dynamics and composition, such as $pCO_2$ (Ludwig et al., 2018; Cao et
al., 2019). The importance of $pCO_2$ during periods of abrupt change, such as the ones that
occurred in the SW IP during the deglaciation, deserves particular attention. Understanding
the temporal dynamics of the regional forest cover, hereafter TMF (Temperate and
Mediterranean Forest), requires exploration ofthe role of different parameters (temperature,
precipitation and $CO_2$). The new multiproxy study of Site U1385 allows the direct comparison
between terrestrial and marine climatic indicators across the LGM and deglaciation at high
(centennial-scale) temporal resolution, and therefore, the detailed reconstruction of abrupt



changes in the vegetation-based atmospheric conditions and SSTs over the SW Iberian margin, as well as its comparison with available Iberian records. The aims of this study are to:

a) Document terrestrial and marine responses to past climate changes at centennial-scale resolution for the LGM and last deglaciation, including the main abrupt events of the last deglaciation (HS1; B-A and YD) at Site U1385A;

b) Explore the main factors driving forest development during the LGM and last deglaciation;

c) Evaluate the evidence for indications of $pCO_2$ thresholds for forest development.

**2. Materials and environmental setting**

**[Figure 1]**

IODP Site U1385 is a composite record of four drillings in the SW Iberian margin (37°34.285′N; 10°7.562′W, 2587 m below sea level - mbsl) located on a spur at the continental slope of the Promontorio dos Principes de Avis, which is elevated above the abyssal plain and free from turbidite influence (Hodell et al., 2015) (Fig. 1). This work focuses on Hole A, a continuous record of 10 corrected revised meter composite depth (crmcd) mainly composed of hemipelagic silt alternating with clay (Hodell et al., 2015). For this study, Hole A was sampled from 3.84 to 1.08 crmcd, which corresponds to the period between 22550 and 6480 cal yr BP. The sediment supply, including pollen grains, to Site 1385 is mainly derived via fluvial transport from the the Tagus and Sado hydrographic basins, providing a reliable signature of the vegetation of the adjacent continent (Naughton et al., 2007; Morales-Molino et al., 2020). The present-day climate of southwestern Iberia is characterised by a Mediterranean climate strongly influenced by the Atlantic Ocean, Köppen classification CSa with warm summers (around 22°C as the average temperature of the warmest month) mean annual temperatures between 12.5°C and 17.5°C, and mean annual precipitation from 400 to 1000 mm/yr. The rainy season peaks in the winter between November and January and drought occurs in the summer generally from June to September.

3. Methods

**3.1. Chronological framework**

**[Table 1, Figure 2, Figure 3, S.M. Fig. 1]**

Eleven AMS [14]C dates were used to generate a new age-model for the last deglaciation at Site U1385 (Table 1 and Fig. 2). Five previously published AMS [14]C dates from Oliveira et al. (2018) (analysed at the Vienna Environmental Research Accelerator (VERA), Isotope Physics Research Group, University of Vienna, Austria, from monospecific foraminifer samples of *Globigerina bulloides*) were also used (Table 1). A new set of six samples for AMS [14]C were selected and processed at the Keck Carbon Cycle AMS Facility, (University of California, Irvine), from monospecific foraminifer samples of *Globigerina bulloides* (Table 1). The new age-model was calculated using a Bayesian approach, through the software Bacon implemented in R (Blaauw and Christen, 2011; R Development Core Team, 2020). We used the Marine13.14d calibration curve (Reimer et al., 2013) which integrates a marine reservoir correction (R) of 500 ± 200 years (Bard et al., 2004a; 2004b; 2013). We calculate a weighted mean DeltaR, based on the ten neighbouring sites (around Site U1385) of 143 ± 139 years, at 1 s.d using CALIB 7.1 (Stuiver et al., 2020) to account for regional effects.

**3.2. Pollen analysis**



A total of 97 samples (including 25 previously published by Oliveira et al., 2018) were analysed
between 3.84 to 1.08 crmcd in Hole A, and prepared at the University of Bordeaux using the
standard protocol of the the UMR EPOC laboratory The sediment was firstly separated using
coarse-sieving at 150 µm, retaining the fine fraction. A sequence of chemical treatments,
starting with cold HCl at increasing concentrations (10%, 25%, 50%) eliminated calcium
carbonate particles. Cold HF. at increasing strength (45% and 70%) eliminated silicates. The
remaining residue was micro-sieved (10 µm mesh), retaining the coarse fraction. Exotic
*Lycopodium* spore tablets of known concentration were added to each sample to calculate
pollen concentrations (Stockmarr, 1971). The obtained residue was mounted in a mobile
medium composed of phenol and glycerol 1% (w/v), to allow the pollen/spore rotation and
accurate identification. Samples were counted using a transmitted light microscope at 400X
and 1000X (oil immersion) magnifications. To perform pollen identification, we used
identification keys (Faegri and Iversen, 1989; Moore et al., 1991), a photographic atlas (Reille,
1992; 1995) and the SW Mediterranean modern reference collection.
The total count ranged from 198 to 1545 pollen and spores per sample, with a minimum of
100 terrestrial pollen grains and 20 pollen morphotypes to provide statistical reliability of the
pollen spectra (McAndrews and King, 1976; Heusser and Balsam, 1977). The main pollen
sum was calculated following previous palynological studies of Site U1385 (e.g. Oliveira et al.,
2016) that excluded *Pinus*, *Cedrus*, aquatic plants, Pteridophyte and other spores, and
indeterminable pollen. The pollen percentages are calculated against the main pollen sum;
*Pinus* and *Cedrus* percentages as well local taxa are calculated against the main pollen sum
plus the taxon. *Pinus* pollen, being an anemophilous taxon, is generally overrepresented in
marine deposits and therefore excluded from the main sum (Naughton et al., 2007). *Cedrus*
being an exotic component possibly transported by wind from the Atlas Mountain (Morocco),
is also excluded. PSIMPOLL 4.27 (Bennett, 2009) was used to plot percentages for selected
taxa, grouped by ecological affinities (Gomes et al., 2020). Stratigraphically constrained
cluster analysis by Sum of Squares (CONISS) determined the five statistically significant
pollen assemblage zones (CONISS) (U1385-1 to 5) based on a dissimilarity matrix of
Euclidean distances with pollen taxa ≥ 1% (Grimm, 1987; Bennet et al., 2009).
**3.3. Compilation of Iberian margin pollen records**
In order to assess vegetation and climate changes in the IP region across the LGM and the
last deglaciation, we compiled available marine records along the Iberian margin covering the
period from 23 to 6 ka. Pollen count datasets from eight pollen records (D13882 - Gomes et
al. 2020; MD03-2697 - Naughton et al., 2016; MD95-2039 – Roucoux et al., 2005; MD95-2043
Fletcher and Sánchez Goñi, 2008; MD95-2042 – Chabaud et al., 2014; ODP Site 976 –
Comborieut Nebout et al., 1998; 2002; 2009; SU81-18 Turon et al., 2003; Site U1385 – this
study) were used with the original published chronologies. Pollen percentages were
recalculated against the main pollen sum. A uniform calculation of the pollen-based ecological
group TMF was made for each record, integrating the following taxa of 1)Temperate trees and
shrubs: deciduous *Quercus, Acer, Betula, Cannabis/Humulus, Carpinus, Castanea, Fraxinus*
*excelsior-type, Hedera helix, Hippophae, Ilex, Juglans, Myrica and Vitis;* and 2) Mediterranean
taxa: evergreen *Quercus, Quercus suber, Arbutus type, Buxus, Daphne, Jasminum,*
*Ligustrum, Myrtus, Olea, Phillyrea, Pistacia, Rhamnus, Rhus.*
To assess the general trend of vegetation patterns throughout the deglaciation, we applied a
Generalised Additive Model (GAM), considered as a more robust statistical approach than





loess curves (Wood, 2017; Simpson, 2018). The GAM model was fitted using the *gam*()
function of the *mgcv* package (version 1.8.24; Wood, 2017) for R (version 3.6.3; R Core Team,
2020). We fitted the model using a standard GAM with REML smoothness selection, with 30
basis functions (*k*=30) and a smoothing parameter of 0.0001 (*sp*=0.0001). To check the
validity of the smooth terms and if the used basis functions captured the wiggliness, we applied
a test using the *gam.check*( ) function of the *mgcv* package. The *k-index* obtained higher than
1, and the *p-value* supported the hypothesis that in both cases, enough basin functions were
used. The curve shows the fitted GAMs for TMF with an approximate 95% confidence interval
(Simpson, 2018).
**3.4. Molecular biomarkers**
Marine biomarker analyses were carried out in 123 levels, including 30 already published by
Oliveira et al. (2018). All analyses were performed following the extraction and analytical
methods described in Villanueva et al. (1997) and Rodrigues et al. (2017).
Marine coccolithophorid algae synthesise organic compounds including alkenones (Volkman
et al., 1980) (Fig. 3i and j). Seawater temperature changes influence the amounts of di-, tri-
and tetra-unsaturated alkenones produced by algae (Brassell et al., 1986). The use of organic
solvents to separate the total lipid fraction from sediments allows the sea surface temperature
alkenone-based reconstruction ($U^{k'}_{37}$ - SST) (e.g. Rodrigues et al., 2017, Villanueva and
Grimalt, 1997), The $U^{k'}_{37}$ index (Prahl and Wakeman, 1987) was converted to temperatures
values using the global calibration equation defined by Müller et al. (1998) with an uncertainty
of 0.5ºC (Grimalt et al., 2001). Additionally, tetra-unsaturated alkenone ($C_{37:4}$) percentages
were calculated due to their potential to identify the occurrence of cold freshwater pulses
associated with iceberg discharges (Bard et al., 2000; Martrat et al., 2007; Rodrigues et al.,
2011, 2017) and therefore, changes in the reorganisation of surface water masses in the North
Atlantic (Rodrigues et al., 2017).
The ratio between $C_{29}$ and $C_{31}$ n-alkanes was also calculated to understand how epicuticular
wax production in terrestrial plants varied through the time (Eglinton and Hamiltom 1967). This
index is generally considered to encompass the dynamic between woody plants vs grasses
plants of the adjacent continent (Cranwell 1973, Tareq et al., 2005, Bush et al., 2013). If the
index is >1, it is typically considered to reflect higher quantities of $C_{29}$ *n*-alkanes by trees and
shrubs, while value of the index <1 are generally considered to indicate the production of
higher quantities of $C_{31}$ *n*-alkanes by grasses and herbaceous plants (Cranwell, 1973; Ortiz et
al., 2010; Rodrigues et al., 2009). This relation encompasses the adaptation of plants, by
increasing leaf wax production, to reduce water loss during the photosynthetic processes and
prevent desiccation promoted by harsh winds or more arid conditions (Bush and McInerney,
2013).

**4. Results and interpretation**
**4.1. Age model**
The studied interval encompasses the period from ~23 to 6 ka, as shown by the radiocarbon
age model (Fig. 2). The average temporal resolution for the pollen and marine biomarkers



across the deglaciation is 127 and 116 years, respectively, or slightly lower (171 and 131
years, respectively) when including the Holocene section (Fig. 3 and S.M. Fig. 1).
**4.2. Major vegetation and climatic shifts in SW Iberia during the last deglaciation**
The U1385 pollen diagram with clustering analysis (SM Fig.1) and SST profile reveals four
main episodes over the LGM and the last deglaciation (Fig. 3, further details in S.M. Table 1).
We emphasise the findings of the new U1385 record but also showcase the excellent
correspondence between the record and the Iberian margin compilation (Fig. 3c), highlighting
generally strong parallels in forest development across the compilation of eight records:
Pollen zone U1385-1 (22550 – 18130 cal yr B.P.) corresponds to the LGM, and shows the
dominance of semi-desertic taxa (STE, ~40%), reflecting dry conditions over the continent
(Fig. 3d). The high values of $C_{29}/C_{31}$ between 0.8 and 1 detected during this interval, might
suggest an increase in epicuticular wax production by woody plants in response to dry
conditions (Fig. 3h). Although STE were the dominant taxa, the moderate presence of
heathland (ERI, ~10-20%) suggests some moisture availability (Fig. 3e) as at present-day they
develop preferentially under oceanic (temperate and moist) climate (e.g. Polunin and Walters,
1985). The low percentages of TMF (5-15%) (Fig. 3c), suggesting cold and dry conditions
over the continent, are consistently observed across the marine records in southerly locations
off the Iberian Peninsula (MD95-2043 - Fletcher and Sánchez Goñi, 2008 and ODP Site 976
- Comborieut Nebout et al., 1998; 2002; 2009 in the Mediterranean Sea, and SU81-18- Turon
et al., 2003 in the Atlantic Ocean) as well as further North off the IP (MD99-2331 and MD03-
2697- Naughton et al., 2007; 2016). This zone is also characterised by moderately cool SSTs
(average ~14.5ºC, Fig. 3j), and minor influence of meltwater pulses as revealed by the low
(not significant, < 2%) $C_{37:4}$ values (Fig. 3i).
Pollen zone U1385-2 (18130 – 15400 cal yr B.P.) corresponds to HS1, and reveals the
maximum expansion of STE (Fig. 3d) suggesting extreme dry conditions over the
southwestern Iberian Peninsula. The decrease observed in more moisture-demanding heaths
(ERI) as well as terrestrial marshes and wetlands (decrease in *Isoetes* undiff.) could be likely
the result of increased moisture stress (Fig. 3e, S.M. Table 1 and S.M. Fig. 1). The high $C_{29}/C_{31}$
values (mostly above 1) observed consistently in this zone suggest a further increase in
epicuticular wax production by the vegetation cover as compared with the preceding LGM
(Fig. 3h). At the same time, the TMF, and especially the thermophilous components, almost
disappeared, confirming dry but also extreme conditions (Fig. 3c and f, S.M. Fig. 1 and S.M.
Table 1). The dominance of STE during HS1 is consistent across the majority of the IP records
(Roucoux et al., 2005; Naughton et al., 2007; 2016; MD95-2043 - Fletcher and Sánchez Goñi,
2008; ODP Site 976 – Comborieut Nebout et al., 2002). In conjunction, SSTs drop to minimum
values (~12ºC, Fig. 3j), reflecting the coldest sea surface conditions of the deglaciation in SW
Iberian margin. The high $C_{37:4}$ values (~8.2%, Fig. 3i) reflect maxima of meltwater pulses,
associated with extreme cold conditions and a clear expression of HS1 in the Atlantic Ocean.
Pollen zone U1385-3 (15400 – 12760 cal yr B.P.) shows a strong development of TMF
including a minor increase in Mediterranean elements(Fig. 3c and f) anda reduction of STE
(Fig. 3d) reflecting a trend of increasing warmth and humidity marking the Bølling-Allerød
episode in the southwestern Iberian Peninsula. Although STE decreases at the onset of this
episode, grasses expand, suggesting still relatively dry conditions during the beginning of this
episode (Fig. 3g and d). The increase of SSTs parallels the terrestrial/atmospheric warming
trend (Fig. 3j and c, although the maximum SST (17.5ºC) was attained before maximum TMF
expansion. The asynchrony, observed at the onset of this zone, between TMF (gradual





increase, Fig. 3c) and SST (maximum values, Fig. 3j) could indicate some moisture deficit at
the start of this zone, and this assumption can be supported not only by the high abundance
of grasses but also by the continued high $C_{29}/C_{31}$ ratio at the onset of this interval (Fig. 3g and
h). Indeed, several other pollen records across IP show a similar pattern of dryness during the
initial phase of the BA (Van der Knaap and van Leeuwen, 1997; Roucoux et al., 2005;
Naughton et al., 2007; 2016; ODP Site 976 – Comborieut Nebout et al., 2002). The most likely
explanation for the delayed response of the TMF is the existence of a moisture deficit at the
start of the BA (Naughton et al., 2016). The rise in Mediterranean elements towards the end
of the zone suggests an increasing expression of warm but dry summers. The high $C_{29}/C_{31}$
ratio at the onset of this zone shifts towards low values by the end of this episode, possibly
reflecting an overall decrease of wax production by plants in response to reduced aridity (Fig.
3h).
Pollen zone U1385-4 (12760 – 11050 cal yr B.P.) corresponds to the YD and initial Holocene.
This zone is marked by a TMF contraction and expansion of STE (Fig. 3c and d), reflecting
regional cooling and drying over the southwestern Iberian Peninsula. There is a slight increase
registered by the $C_{29}/C_{31}$ ratio, consistent with the patterns observed in preceding zones which
could again be associated with an increase in the leaf wax production under more arid
conditions (Fig. 3h). A significant SST reduction is observed with a minimum of 13.2 ºC in the
record (Fig. 3j). However, in contrast to HS1, freshwater pulses are insignificant during this
phase (Fig. 3i). The fairly weak reduction in TMF observed inin our record and corroborated
by the compiled records (Fig 3c) contrasts with the steppe environment described for this
interval, especially in the southeast of the IP (Carrión et al., 2002; Camuera et al., 2019). A
more pronounced forest contraction is observed in the high altitude terrestrial/lacustrine cores
(Quintanar de la Sierra II – Peñalba et al., 1997; and La Roya - Allen et al., 1996) in which the
near-disappearance of the forest might reflect the altitudinal adjustments in vegetation belts
(Aranbarri et al., 2014). However, the U1385 record and other Iberian margin and IP records
(e.g. Lake de Banyoles – Perez-Obiol and Julià, 1994; MD03-2697 – Naughton et al., 2007;
MD95-2039 – Roucoux et al., 2005; Charco da Candieira – van der Knaap and van Leeuwen,
1997; MD95-2042 – Chabaud et al., 2014; D13882 - Naughton et al., 2019; MD95-2043-
Fletcher and Sánchez Goñi, 2008; ODP Site 976 – Comborieut Nebout et al., 2002) show a
relatively high percentage of TMF during the YD when compared with the previous HS1 in the
SW IP (Fig. 3c).
Pollen zone U1385-5 (11050 – 4500 cal yr B.P.) corresponds to the early to Middle Holocene.
This zone is marked by the expansion of TMF as well as the Mediterranean forest, reflecting
a regional increase in temperature and precipitation. Despite the low temporal resolution it is
consistent with nearby records with a maximum forest development at around 9000 cal yr B.
P. (Gomes et al., 2020). Minimum values of $C_{29}/C_{31}$ ratio suggested a decrease in epicuticular
wax production by vegetation possibly do to the most favourable conditions for vegetation
development compared with the preceding zones.During this zone is noethworthy the warmer
SST around 18-20 ºC.
**5. Discussion**
**[Figure 4], [Figure 5]**
**5.1. The effect of pCO$_2$ on biome changes during the LGM and deglaciation**
Whilst a classic interpretation of ecosystem dynamics as described for Site U1385 can be
proposed solely considering the variation of the main climatic parameters (temperature,





precipitation), we hypothesise that past changes in $pCO_2$ played an essential role in vegetation
change, specifically in the deglacial forest expansion. Here we re-evaluate the drivers of
vegetation change, explicitly considering the evolution of $pCO_2$ through the deglaciation. Our
discussion is informed by the present-day environmental and climatic space occupied by
different taxa in Portugal (Temperate Mediterranean forest – *Quercus* sp., Heathland -
Ericaceae family and semi-desertic taxa) (S.M. Fig. 2).

**LGM**

The pollen-based vegetation record from Site U1385 shows that during the LGM a grassland-
heathland mosaic dominated the landscape (Fig. 3d, e and Fig. 4d), a distinctive non-analogue
glacial vegetation cover. The prevalence of heath pollen in Iberian pollen records underpins
the classic view of the LGM in Iberia as a fairly humid interval, certainly compared with the
extreme aridity evident during Heinrich stadials (Naughton et al. 2007; Roucoux et al. 2005;
Sanchez-Goni et al. 2009; Combourieu-Nebout et al. 2009; Fletcher and Sanchez-Goni 2008).
Nevertheless the juxtaposition of high abundances of semi-desert and heathland taxa remains
intriguing. In terms of their present-day climatic space distribution, the STE and ERI taxa differ
in that the latter can occupy niches with high humidity, which contrasts with the arid-loving
conditions of the former (S.M. Fig. 2c). Interestingly, the environmental space for the
Ericaceae group (namely *Erica arborea*, *E. australis*, *Calluna vulgaris*) coincides with that
occupied by the *Quercus* genus, the main constituent of the TMF group (S.M. Fig. 2b). This
begs the question, if the environmental conditions that support heathland overlap with those
for *Quercus* sp., then why were forests not thriving during the LGM? The first answer could be
cold atmospheric temperatures, even if during the LGM the temperatures were not as extreme
as the ones observed during the HS1 (Bond et al., 1993; Rasmussen et al., 1996). As such a
potential controlling factor could be the low levels of $pCO_2$ during the LGM of between 180-
190 ppmv, which is amongst the lowest concentrations recorded during the history of land
plants (Pearson and Palmer, 2000; Tripati et al., 2009). The global distribution of different
vegetation types as a function of temperature and precipitation was modelled under modern
and corrected for LGM $CO_2$ (185 ppm) showing qualitative differences in the distribution of
vegetation types (Shao et al., 2018). Under low $pCO_2$ grassland was favoured to the detriment
of evergreen broadleaf, evergreen and deciduous needle leaf forest. That study, however, did
not include ericaceous heathlands specifically, and it is not known whether this group has
adaptations permitting better functioning under low $pCO_2$ levels. We speculate that drought-
adapted traits in Mediterranean Ericaceae especially *E. arborea* including thick cuticles, small
leaf size, large photosynthetic thermal window and deep root system with large diameter and
a massive underground lignotuber (Gratani and Varone, 2004) may have been beneficial in
coping with the challenging trade-off between photosynthesis and water loss under very low
$pCO_2$. As such, the Ericaceae of the LGM may represent part of vegetation that coped well
with physiological constraints of the low $pCO_2$ world.
At the same time, we notethatthe LGM correpsonds to  a maximum in of the precession cycle,
which is recognised to promote a weakening of seasonal contrasts(reduced summer dryness)
and consistently associated with heathland development in the Iberian Peninsula (Fletcher
and Sanchez Goni, 2008; Sanchez Goni, 2008), in both glacials and interglacials (e.g. Oliveira
et al., 2017), including the Middle to Late Holocene (Gomes et al., 2020; Oliveira et al., 2018;
Chabaud et al., 2014). As such, during the LGM, the precession maximum promoting wetter
summers should have been a trigger for heathland development .





Diverse vegetation models have been used to understand the influence of climatic parameters
and $pCO_2$ during the LGM (e.g. Harrison and Prentice, 2003; Woillez et al., 2011; Shao et al.,
2018). There is a disagreement about the magnitude of the $pCO_2$ influence, from being
considered to have an equal influence (Izumi and Lezine, 2016) to being thought to be less
critical than climatic parameters (Woillez et al., 2011; Shao et al., 2018; Chen et al., 2019).
Harrison and Prentice (2003) also highlight models differences and the variable regional
expression of the influence of $pCO_2$ (with higher impact in tropical areas). However, these
studies agree that low $pCO_2$ had a negative physiological impact on forest development during
the LGM in different continents (Jolly and Haxeltine, 1997; Cowling, 1999; Harrison and
Prentice, 2003; Woillez et al., 2011; Shao et al., 2018; Chen et al., 2019). Jolly and Haxeltine
(1997) used BIOMOD to simulate LGM vs pre-industrial $CO_2$ levels under different climatic
conditions scenarios (temperature and precipitation) in tropical Africa; $CO_2$ was considered
the primary driver of biome change from tropical montane forests to shrubby heathland
ecosystems. This model included a photosynthetic scheme able to simulate plant response to
different levels of $CO_2$ and its impact on stomatal conductance and water stress. This study
showed that increasing $pCO_2$ (above ~190 ppmv), offsets the lower temperatures (changes of
-4 to -6 °C), allowing the forest to thrive and substitute heathland. However, plants with higher
climatic demands (temperature and precipitation), which is the case of most temperate trees,
are less competitive under low $pCO_2$ conditions, compared with evergreen microphyllous
species (e.g. *Erica* sp.). The ecological advantages of *Erica* sp. also include less demanding
edaphic requirements (low nutrient demand), more competitive re-sprouting strategy after
disturbance, especially fires, as well as a higher dispersal capacity compared with *Quercus*
sp. for example (Pausas, 2008).
The inclusion of $pCO_2$ in climatic reconstructions for LGM for Africa and Europe yields a wetter
LGM compared with reconstructions assuming $pCO_2$ present-day concentrations (Wu et al.,
2007). The implications of these experiments are important for the southwest Iberian region
and may help to resolve the apparent contradiction between vegetation (abundance of semi-
desertic plants and presence of heathland) and climate simulations which indicate enhanced
winter precipitation over southern Iberian and Northwest Africa due to southward shifting of
the wintertime westerlies (Beghin et al., 2016). In the absence of $pCO_2$ correction, temperature
could also be misinterpreted; the LGM vegetation for Mediterranean sites was simulated and
associated with warmer summer under LGM $pCO_2$, instead of the more cold conditions
simulated with present-day levels of $CO_2$ (Guiot et al., 2000). In Europe, pollen reconstruction
with steppe vegetation indicated warmer winter temperature for LGM $pCO_2$ compared with the
modern $pCO_2$ (Wu et al., 2007). The bias could extend to vegetation reconstructions; without
the $pCO_2$ effect, the cover of boreal and temperate forests is reduced, and evergreen forests
are overestimated for the LGM (Woillez et al., 2011).
Experiments determining plant thresholds in response to low $pCO_2$ have not received as much
attention as research on the impact of high $pCO_2$ levels (Gerhart and Ward, 2010; Dusenge
et al., 2019). When we assess the relationship between $pCO_2$, SST and TMF across the LGM
and deglaciation events we observe that LGM occurrence of TMF (i) corresponds to SSTs
below 15.5°C and $pCO_2$ below 225 ppmv and (ii) that values remain below 20% (Fig. 5). Within
African mountain environments, a value of 220 ppmv of $pCO_2$ has been suggested as a
threshold above which the forest could develop (Dupont e al, 2019). Therefore, we can infer
that the forest development In SW of Iberian during LGM may have been constrained by the
interaction of relatively low temperatures (with seasonal oscillations)and low levels of $pCO_2$
(~185 ppmv). One could speculate that a hypothetical increase of $pCO_2$, above 225 ppmv
values during the LGM would have permitted forest development in the southwest Iberian





Peninsula, although independent proxies for terrestrial temperatures and precipitation amount
are highly desirable.
**HS1**
During HS1, a Mediterranean steppe landscape with minimum arboreal development resulted
from the lowest temperatures and highest levels of aridity observed within the studied interval
(Fig. 3 and 5c). During this event, the potential effect of increasing $pCO_2$ (from ~185 to ~225
ppm) from 18.1 to ~16 cal ka B.P. (Fig. 3b) was not enough to counteract the limiting effect of
the climate conditions (coldest and driest atmospheric conditions), and indeed should have
exacerbated aridity stress at this time.. Regional models (Weather and Research Forecast
Model – WRF) reconstructing the potential vegetation with a $pCO_2$ correction show a reduction
in arboreal vegetation and increase of sparsely vegetated soil for the IP region during HS1
compared with the LGM (Ludwig et al., 2018). The reconstructed precipitation values for the
southwestern Iberian Peninsual (Tagus hydrographic basin catchment), show values below
700 mm/yr for HS1, which agrees with the pollen data and again the widespread semi-desertic
taxa development. Interestingly, the differences between HS1 and LGM are quite relevant,
which could explain the stronger development of the heathland in the LGM. The reconstructed
atmospheric temperature showed a longitudinal variation between the HS1 and LGM varying
from -2.5 to -1.5 °C; which are in line with the high percentages of semi-desertic taxa of Site
U1385, as well as other nearby IP records (Peñalba et al., 1997; Perez-Obiol and Julia, 1984;
Comborieu Nebout et al., 2002; Roucoux et al., 2005; Naughton et al., 2007; Fletcher and
Sánchez Goñi, 2008). Besides, the forest development was constrained across the territory,
and based on pollen data from marine and terrestrial records we do not observe any significant
(<5% TMF) latitudinal difference when comparing northern (e.g. Peñalba et al., 1997; Perez-
Obiol and Julia, 1984; Roucoux et al., 2005; Naughton et al., 2007) with southern (e.g. this
study; Comborieu Nebout et al., 2002; Fletcher and Sánchez Goñi, 2008) pollen records.
Furthermore, the relationship between $pCO_2$, SST and TMF across the HS1 show scattered
values of TMF (below 20%) occurring at temperatures below 15.5°C and $pCO_2$ below 225
ppmv (Fig. 5).
**BA**
The BA is characterised by favourable climatic conditions (higher temperatures, higher
moisture availability) for TMF development (Fig. 3c). The high temperature and a dry to wet
trend are likely the primary drivers of progressive forest development during the BA. However,
it is important also to consider a possible role of increasing of soil maturation (higher organic
matter content, pH, plant nutrients, during succession/development of this biome), as well as
a possible "fertilisation effect" of the stepwise increases of $pCO_2$ by ~15 ppmv around this time
interval (Fig. 3b). The simulations produced by BIOME3 for African Biomes (Tropical
forest/Ericaceous scrub) with a present climate showed that above 190 ppmv, the increase of
$pCO_2$ at intervals <20 ppmv, gradually offsets the negative effect of temperature changes;
above 250 ppmv with a maximum temperature change of ~-6°C the development of forest will
be promoted in detriment of the ericaceous scrubland (Jolly and Haxeltine, 1997).
The abrupt increases in $pCO_2$ at 16.3 Ka and 14.8 Ka (Marcott et al., 2014) (Fig. 3b), could
tentatively be associated with the slight increase of forest at the onset of the BA and with the
highest peaks of forest development observed during the BA (within age uncertainties of both
archives) (Fig. 3c). Cao et al. (2019), using pollen-based biome reconstruction, suggested that



worldwide expansion of forests was a consequence of the increasing $pCO_2$ superimposed
over the temperature increase between 21 ka and 14 ka. Cao et al. (2019) further emphasise
the role of $CO_2$ after the LGM driving a general northward expansion of forests and
replacement of grassland by temperate forests in Europe, by minimising moisture limitation
and enhancing WUE. Afterwards, from 14 ka to 9 ka, sufficient moisture (in a general
assumption) might also have played a significant role(Cao et al., 2019), whilst the higher levels
of $pCO_2$ may have been able to offset, at least at the end of the BA, the effect of any potential
reductions in moisture availability. During the BA, considering that temperature and moisture
availability was quasi-optimal, increases in $pCO_2$ levels (>225 ppmv) may have amplified TMF
expansion during this period (Fig. 4b and Fig. 5).
**YD**
The YD is characterised by a forest-grassland mosaic, as indicated by relatively high levels of
forest coexisting with semi-desertic taxa (Fig. 3c, d and Fig. 4a). Strong SST cooling (Fig. 3j)
may have been associated with cooler land surface temperatures. However, this impact may
have been offsetby the positive effect of sufficient moisture availability (based on the presence
of TMF, Naughton et al. 2019) and the increasing trend of $pCO_2$ (Fig. 3b). Unfortunately, there
is a lack of independent precipitation proxies for this region, and Dennison et al. (2018)
highlight a lack of reliability in the speleothem proxies for precipitation in this region for this
time interval. We observe that the YD forest development occurs in association with similar
SSTs to those of the LGM and slightly higher than those of HS1. Meanwhile, $pCO_2$ was above
the 225 ppmv thresholdthroughout the YD, (reaching maximum values of ~260 ppmv, at ~12
Ka) (Fig. 5). The increase in $pCO_2$ may have enhanced plant productivity and WUE (Cowling
and Sykes, 1999; Ward et al., 2005) during the YD, partially compensating for the impact of
atmospheric cooling and drying. Schenk et al. (2018) suggest $pCO_2$ may play an essential role
in the forest development if enough moisture is available. It may be that the tree cover was
restricted to suitable, moist microhabitats and close to refuge zones, but was not as restricted
as in previous cold periods (Svenning et al., 2011), as pollen data also suggests (Fig. 3c).
Also, simulated data from vegetation-climate models based on pollen records for biome
reconstruction (Shao et al., 2018) and in a dynamic vegetation model (ORCHIDEE) driven by
outputs from an AOGCM (Woillez et al., 2011) emphasise the influence of increasing $pCO_2$ as
a critical factor for worldwide forest development during the period including the YD (Shao et
al., 2018). Underlying these changes the increase in summer insolation (Fig. 3a), which
contributed to the increase of summer temperatures cannot be neglected as a promotor of
forest development, at least where trees where not excessively water-stressed. However
disentangling the isolated contribution of insolation vs $pCO_2$ requires sensitivity experiments,
not yet performed. In summary, the persistence of TMF during the YD, despite cold
temperatures with some seasonality, (warmer than the HS1), seems to be best explained by
the combined interaction between sufficient moisture availability, higher atmospheric
temperature, at least during summer (promoting forest development) and increasing $pCO_2$
(between ~245 and 265 ppmv) (Fig. 4a).
**5.2. $C_{29}/C_{31}$ ratio and $C_3/C_4$ dynamics: potentials and limitations**
Insights into the dominance of different plant physiological pathways can be potentially gained
using $C_{29}/C_{31}$ n-alkanes of Site U1385A (Fig. 3h). In general, $C_{29}$ and $C_{31}$, as well as other long-
chain alkanes with odd carbon numbers (e.g. $C_{29}$, $C_{31}$, $C_{33}$), are epicuticular waxes produced



by terrestrial plants, from which $C_{29}$ could represent woody plants and $C_{31}$ grasses (Meyers,
2003). However, caution in interpreting the $C_{29}/C_{31}$ ratio in terms of taxonomic groups is
required since woody plants and grasses are both capable of producing $C_{29}$ and $C_{31}$ chain
lengths (Ortiz et al., 2010; Bush and McInerney, 2013). Furthermore, regional differences are
observed across the world and between biomes in terms of what long-chain n-alkanes a
species is producing (Bush and McInerney, 2013). Noting this limitation, the analysis of $C_{29}/C_{31}$
curve shows increasing values during the LGM to yield high values during the HS1, followed
by the YD underlying a decreasing trend towards the Holocene (Fig. 3h). The $C_{29}/C_{31}$ is
positively: r = 52% (negatively: r = -63%) correlated (Pearson's correlation coefficient) with the
semi-desertic (temperate Mediterranean forest) signals within this region over the same
interval (Fig. 3c, d and h). Therefore, we note that the anticipated general interpretation of the
$C_{29}/C_{31}$ ratio as an indicator of the relative abundance of trees vs grasses does not hold for
our datasets (indeed the reverse is evident). Instead, we tentatively infer that $C_{29}/C_{31}$ ratio in
this setting is expressing an adaptation of plants to aridity, and perhaps an increase in wind
strength conditions, which alter the moisture balance. The n-alkanes of leaf waxes are
produced to protect plants against the loss of water during the photosynthetic process (Post-
Beittenmiller, 1996; Jetter et al., 2006). We could expect that arid/cold conditions to be more
disturbing for woody plants than for grasses, as such the increase of the $C_{29}/C_{31}$ during HS1
and YD, could suggest a climatic adaptation of woody plants by increasing the production of
leaf wax $C_{29}$. However, the traditional taxonomic generalisation of $C_{29}$ woody versus $C_{31}$
grasses (Meyers, 2003), still needs some caution.
Other hypotheses to be explored and understood include the connection between the long-
chain n-alkanes and the dynamic between $C_3$ and $C_4$ plants. Nowadays, African savannahs
are dominated by $C_4$ plants, and biomarkers (including $C_{31}$ n-alkanes) can be used to infer
their presence in past landscapes (Dupont et al., 2019). Worldwide, 80% of Poaceae (grasses)
and Cyperaceae (sedges) present a $C_4$ photosynthetic pathway (Sage, 2017) but with pollen
analysis, there is no confidence about the Poaceae and Cyperaceae pollen morphologic types
being exclusively or in its majority $C_4$ plants. We have grouped the Poaceae and the
Cyperaceae pollen taxa, noting the inherent limitations of this grouping (Fig. 3g). This group
(Poaceae + Cyperaceae) presents relatively high values with considerable oscillations
(potentially related to differences in time resolution) between the LGM and the BA and more
stable behaviour onwards. No particular correlation with other indicators (TMF or STE or
$C_{29}/C_{31}$) was evident, apart from the apparent instability before the Holocene. Interestingly,
within a laboratory setting, $C_3$ grasses are favoured in comparison with $C_4$ grasses, when
temperatures increase by 5 to 15˚C with a $pCO_2$ of 200 ppm (Ehleringer et al., 1997; Edwards
et al., 2010). Furthermore, $C_4$ plants nowadays are mostly confined to the tropical grasslands
and savannahs; they are better adapted to environments with higher temperatures, aridity,
poor nutrient soils, and intensive disturbance caused by animals or fire regimes (Bond et al.,
2005; Edwards et al., 2010). Likewise, one should expect that in the Iberia after the LGM (Fig.
3 and 5) should be mainly composed by $C_3$ plants; considering the estimated SSTs indicating
relatively cold temperatures (Fig. 5) and the high percentages of *Artemisia* ($C_3$ plant) (S.M.
Fig. 1).
However, it is not currently possible to entirely rule out an increased importance of $C_4$ plants
in the glacial vegetation in the IP, because pollen morphology does not allow the separation
of these groups and biomarkers proxies have not been tested or reported to clarify the dynamic
between $C_3/C_4$ plants in the Temperate/Mediterranean biomes. The discrimination of $C_3/C_4$
grasses has been made on the basis of stable isotopes of ancient grass pollen (Nelson et al.,
2016) although the single grain isotopic measurements employed remain challenging to





implement. This highlights the theoretical possibility of the $C_3/C_4$ plant dynamic observed in
Africa (e.g. Dupont et al., 2019) and other savannahs ecosystems not being replicable (with
the current knowledge) in our study area. Biomarker species/groups fingerprinting studies are
required in order to eventually distinguish between $C_3$ and $C_4$ plants and then go onto exploring
the dynamics observed between $C_3$ and $C_4$, within IP-Mediterranean ecosystems during the
last deglaciation.

**6. Conclusion**

This study presents high-resolution pollen and SST records from Site U1385 which can be
used in future regional and global reconstructions and models, especially for the Iberian
Peninsula. A long-term analysis of climatic changes was comparable and consistent across
the Iberian records analysed, with the advantage of the new record having an average higher
resolution and a more robust radiocarbon chronology.
We explore the understanding of TMF dynamics under the influence of climatic change and
increasing $pCO_2$ throughout the LGM and deglaciation. Our analysis suggests that forest
development during the LGM may have been also constrained at least in part due to the low
$pCO_2$, acting as a modulator. The baseline climatic conditions to support heathland
development at present in the region are relatively similar to the ones required by some
*Quercus* sp, however, trees development benefits from more warmth months (Polunin and
Walters, 1984). During the LGM, the associated cold conditions and low seasonality together
with the exacerbation of drought stress resulting from the low concentration of $pCO_2$ might
have limited forest expansion. We speculate that certain traits of the Mediterranean Ericaceae,
including small leaf size, thick cuticular waxes and deep rooting which contribute to drought
tolerance at present may have promoted the development of heathlands during the LGM, as
previously observed in African uplands. During HS1, woody plant development was further
restricted by the impact of low temperatures as well aridity, under low $pCO_2$ and associated
with wider climatic perturbation evidenced in freshwater pulses. The BA characterises the
most suitable conditions for TMF development – warm, rising temperatures, moisture
availability, amplification of seasonality, and the increase of $pCO_2$. The TMF persistence, and
the forest–grassland mosaic, during the YD, can be best explained by the joint imprint of
moisture availability and higher $pCO_2$. The role of $pCO_2$ was, in our opinion, fundamental for
the significant TMF development during the late glacial in southwestern Iberia, by comparison
with precedent cold intervals (LGM and HS1). Although other co-hypotheses must be better
assessed, ideally against future development of independent (non-vegetation) proxies for
precipitation and temperature during this time-slice, so far there are no regional
reconstructions that consider the co-effect of moisture and $pCO_2$.
Considering the response of TMF and xerophytic taxa in our pollen record, we consider the
$pCO_2$ value of ~225 ppmas a critical limit for forest expansion in the IP during the last
deglaciation. This hypothesis should be explored through model simulations to establish the
amplitude and critical thresholds of $pCO_2$ impacts on regional vegetation, as well as, in past
cold periods.
The relation of $C_3$ and $C_4$ plants in the Mediterranean domain needs further attention since the
long-chain *n*-alkanes do not yet provide a reliable picture to disentangle the dynamic between
woody plants and grasses. We applied a biomarker proxy $C_{29}/C_{31}$ which is positively correlated
with the semi-desertic pollen curve and negatively with TMF. This points to its potential as a
proxy of aridity, testifying the increase of leaf-wax $C_{29}$ production during the dry periods, albeit
in a regionally-specific way, and noting that this is not in agreement with previous inferences




regarding the discrimination of herbaceous and arboreal taxa. Another suggestion is to test
the $C_{29}/C_{31}$ ratio for other periods in the past, throughout glacial periods.
Many global-scale LGM and deglacial reconstructions have been undertakenwith a
preferential focus on the LGM and YD. An enhanced effort by the modelling community in
developing transient regional simulations covering the last deglaciation may be valuable, to
allow a more precise comparison/testing with proxy data. Our new data and regional pollen
synthesis provide a good target for modelling. Furthermore, this study can provide a baseline
understanding and essential context (potential analogue) for present-day world changes in
arid and semi-arid ecosystems in terms of their potential future evolution under rapidly
changing $pCO_2$.

**Author contribution**

SDG, WF, FN and AS contributed to the conception and design of the study, data analysis
and interpretation. Also they were responsible for the grant application to NERC. SDG
performed pollen analysis. TR performed biomarkers analysis. AR perfomed assemblage
foraminifers picking for radiocarbon dating and draw figure 1. SDG prepared the original draft
and wrote the manuscript including figures with the critical input (edition and revision) from all
co-authors.

**Competing interests**

The authors declare that they have no conflict of interest.

**Acknowledgements**

This research was supported by the Portuguese Foundation for Science and Technology
(FCT) SFRH/BD/128984/2017 PhD grant to SDG, the ULTImATum (IF/01489/2015) and, the
Hydroshifts (PTDC/CTA-CLI/4297/2021) projects; CCMAR FCT Research Unit - project
UIDB/04326/2020, CCMAR BCC grant (Incentivo/MAR/LA00015/2014) to FN, FCT contract
(CEECIND/02208/2017) to DO, WarmWorld Project (PTDC/CTA-GEO/29897/2017) for
Biomarker analyses, and grant (SFRH/BPD/108600/2015) to TR. The six radiocarbon dates
were obtained through the NERC radiocarbon application award NERC 2136.1018. The
contributions of L. Devaux are gratefully acknowledged (Bordeaux 1 University, EPOC, UMR-
CNRS 5805) for his assistance in palynological treatments.

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



**Tables  and figures**
**Table 1 –** Radiocarbon ages of IODP Site U1385.

| Lab code | Core Depth (crmcd) | Material | Conv. AMS $^{14}$C (yr B.P.) | Error | Weighted mean deltaR |
|---|---|---|---|---|---|
| *20140801r9_MSGforam01_5ox | 52 | *G. bulloides* | 2525 | 28 | 143 |
| *20140801r5_MSGforam01_1ox | 108 | *G. bulloides* | 6181 | 35 | 143 |
| *20140801r6_MSGforam01_2ox | 156 | *G. bulloides* | 10060 | 33 | 143 |
| UCIAMS-219300 | 186 | *G. bulloides* | 11310 | 60 | 143 |
| *20140801r7_MSGforam01_3ox | 193 | *G. bulloides* | 11499 | 43 | 143 |
| UCIAMS-219301 | 217 | *G. bulloides* | 12300 | 40 | 143 |
| UCIAMS-219302 | 237 | *G. bulloides* | 13430 | 110 | 143 |
| *20140801r8_MSGforam01_4ox | 246 | *G. bulloides* | 13355 | 45 | 143 |
| UCIAMS-219303 | 251 | *G. bulloides* | 13670 | 60 | 143 |
| UCIAMS-219304 | 303 | *G. bulloides* | 15890 | 70 | 143 |
| UCIAMS-219305 | 333 | *G. bulloides* | 17090 | 90 | 143 |

* AMS from Oliveira et al. (2018)





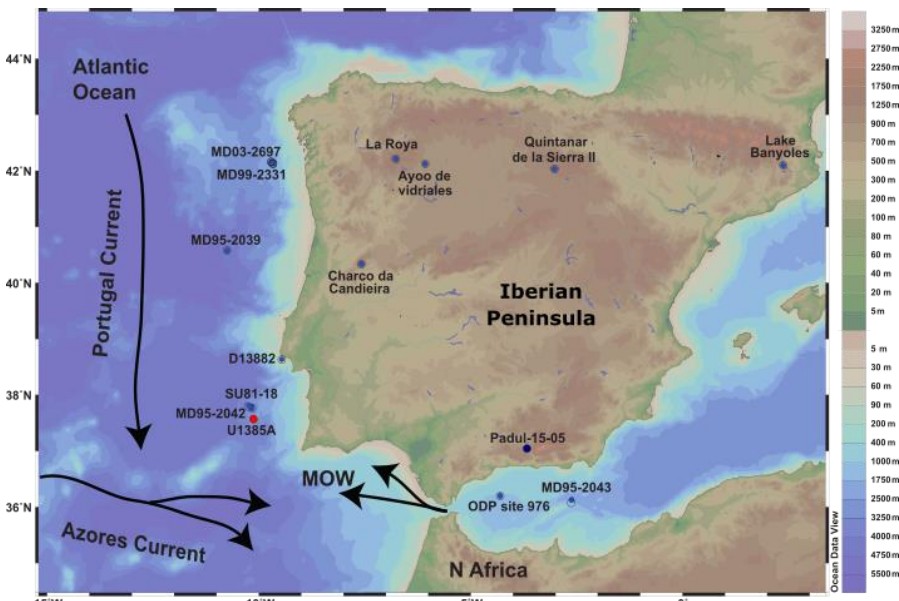


**Figure 1 –** Location of the IODP Site U1385 and of the marine and terrestrial pollen records
discussed in the text. Marine sedimentary records: MD03-2697 (Naughton et al., 2016); MD99-
2331 (Naughton et al., 2007); MD95-2039 (Roucoux et al., 2005); D13882 (Gomes et al.,
2020); MD95-2043 (Fletcher and Sánchez Goñi, 2008); MD95-2042 (Chabaud et al., 2014);
SU81-18 (Turon et al., 2003); ODP Site 976 (Comborieut Nebout et al., 1998; 2002; 2009);
Continental sedimentary records: Lake de Banyoles (Pèrez-Obiol and Julià,1994); Quintanar
de la Sierra II (Peñalba et al., 1997); La Roya (Allen et al., 1996); Ayoo de vidriales (Morales-
Molino and Garcia-Anton, 2014); Charco da Candieira (Van der Knaap and van Leeuwen,
1997); Padul15-05 (Camuera et al., 2019). Black arrows represent the surface water
circulation (MOW, Portugal and Azores Current). Note that coastline boundaries are for the
present day.












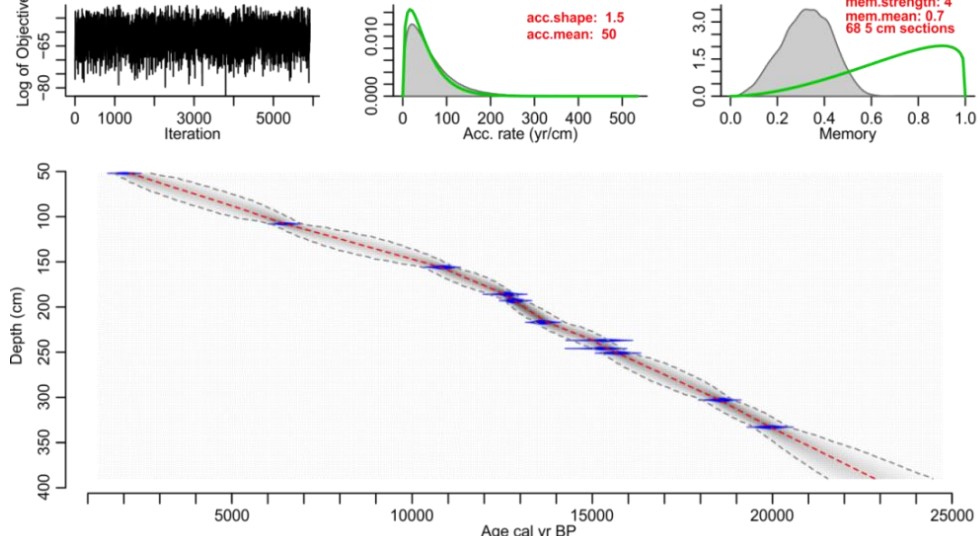


**Figure 2 –** Age-depth model for IODP Site U1385 using a Bayesian approach with Bacon
v.2.3.5 (Blaauw and Christen, 2011). The original dates were calibrated using Marine 13.14c
(Reimer et al., 2013) grey stippled line show 95% confidence intervals; red curve shows single
"best" model based on the mean age for each depth. Upper graphs show from left to right:
Markov Chain Monte Carlo (MCMC) iterations and priors (green line) and posteriors (dark grey
line with a grey fill) for the accumulation rate and variability/memory. Note: the depth (Y axis)
was converted to cm from the corrected revised meter composite depth (crmcd).



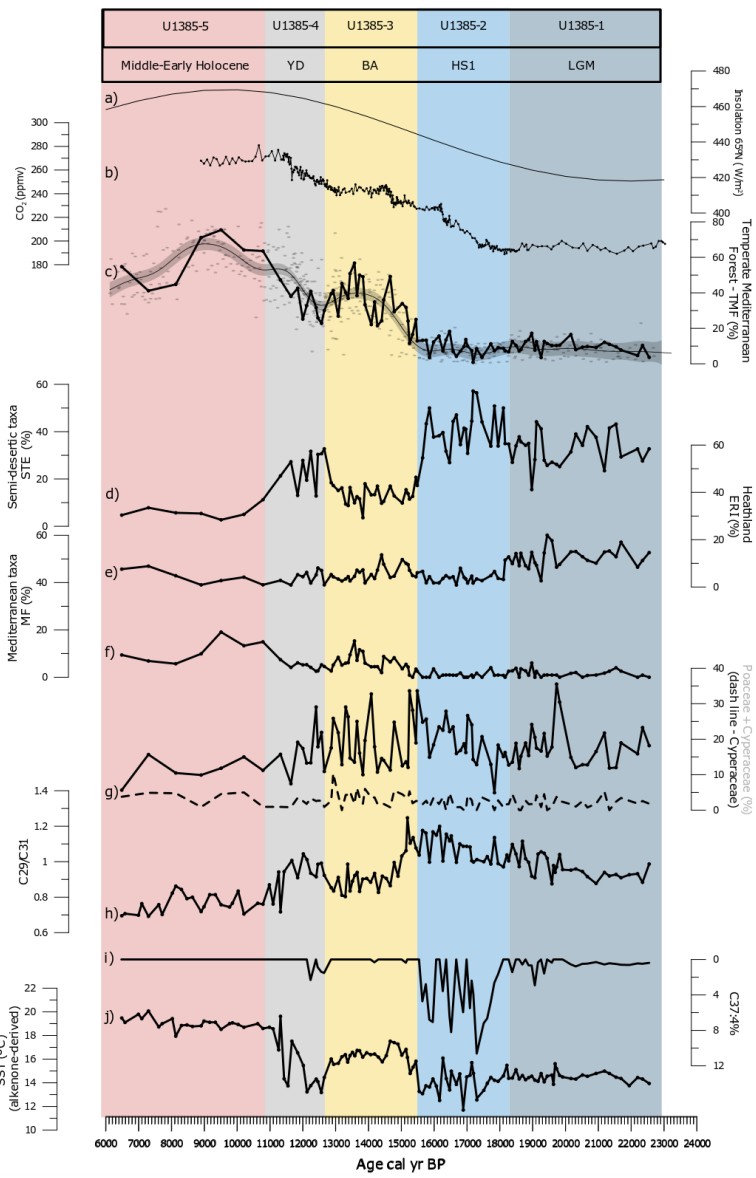


**Figure 3 –** Comparison of multiproxy records from the Site U1385 with 65˚N July (W/m$^2$) summer insolation (Berger and Loutre, 1991) and CO2 composite from WAIS (Marcott et al., 2014) ; b) $CO_2$ (ppmv); Principal pollen-based ecological groups: c) Temperate Mediterranean Forest (%) (solid black line) and Compilation of Iberian Margin TMF records (D13882, MD03-2697; MD95-2042; MD95-2043; ODP-976; U1385) – GAM (curve with grey (%), d) Semi-desertic taxa (%), e) Heathland (%), f) Mediterranean taxa (%) and g) Poaceae + Cyperaceae (%, dash line); h) $C_{29}/C_{31}$ ratio, i) $C_{37:4}$ (%) and j) SST (°C). The different coloured shading corresponds to the pollen zones (SM Fig.1 and S.M. Table 1) and were connected with the periods indicated.



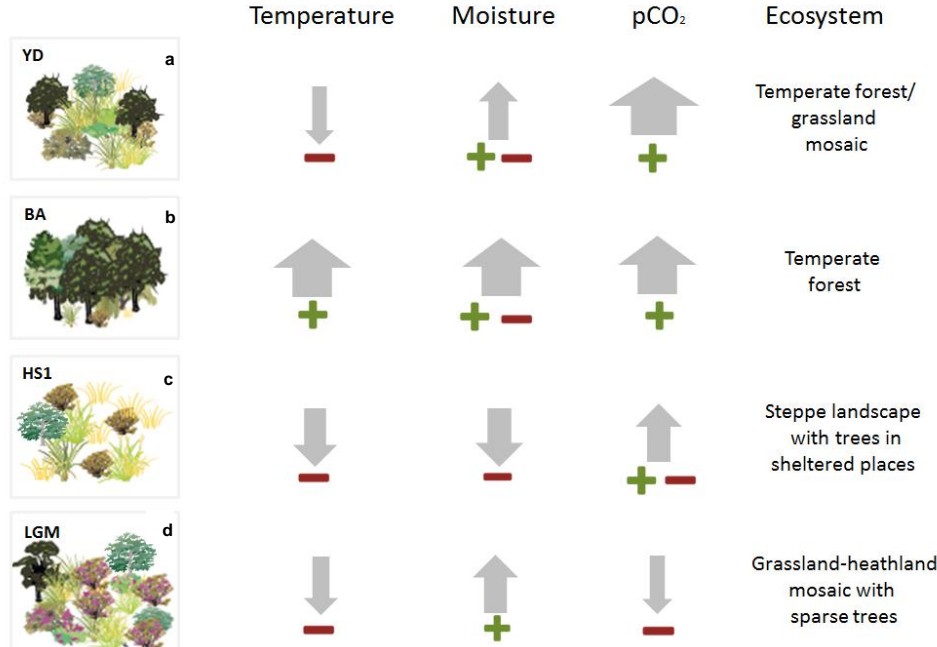

**Figure 4 –** Schematic representation of the influence of climatic parameters (precipitation and temperature) as well as the physiological contribution of CO2 for each period showing a schematic reconstruction of the ecosystem scenarios.



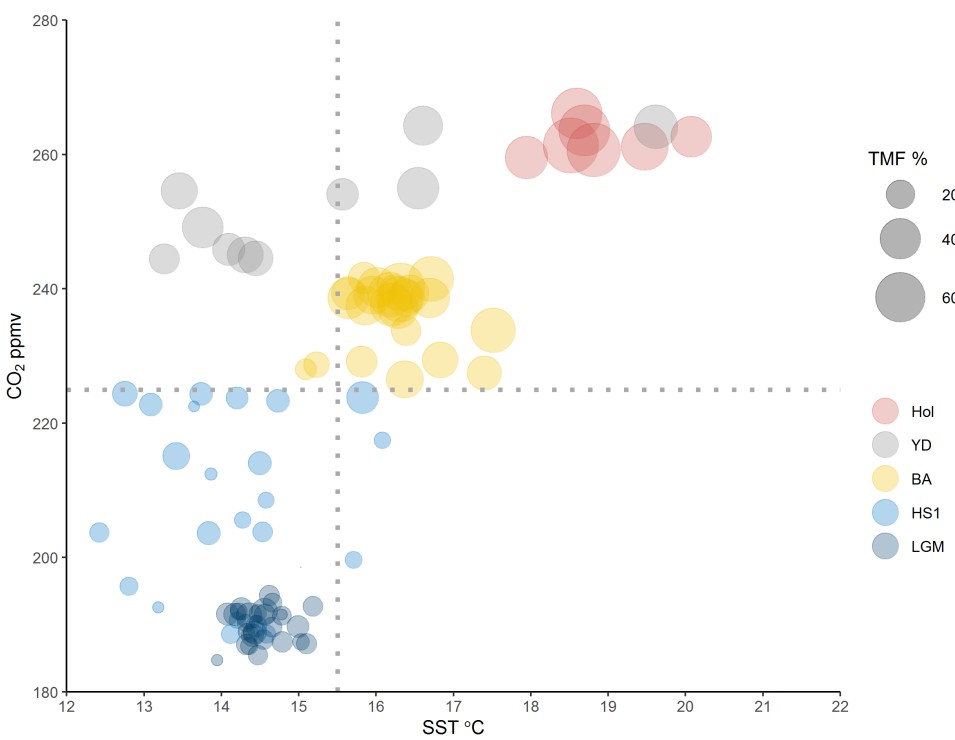


**Figure 5** – Dispersion plot showing the relation between $CO_2$ (Marcott et al., 2014) and SST
in relation to TMF % across the different intervals of the deglaciation, following the pollen
zones boundaries.