# Peer review of "Rising atmospheric CO2 concentrations: the overlooked factor promoting SW Iberian"

_EGUsphere, 2024_

## Referee Comment (RC1)

**Review of "Rising atmospheric CO2 concentrations: the overlooked factor promoting SW Iberian Forest development across the LGM and the last deglaciation?"**

The manuscript by Gomes et al. presents a new high-resolution pollen record for the last deglaciation from core U1385 on the Iberian Margin. Based on the palynological data, non-pollen proxies from the same core, and published marine and terrestrial pollen records, the authors investigate the role of temperature, moisture, and $CO_2$ for the development of forests on the Iberian Peninsula. The manuscript proposes that $CO_2$ plays a bigger role than previously thought, in particular promoting the existence of forests during the Younger Dryas while preventing forest development during the Last Glacial Maximum and Heinrich Stadial 1. The analysis is sound and the conclusions are supported by the presented data, although the wording and clarity of the manuscript could be improved. Therefore, I believe that the manuscript could be suitable for publication in Biogeosciences after addressing the comments below.

**General comments**

- While the topic of the manuscript is suitable for publication in BG, the paper is rather difficult to read without a strong paleoclimate background due to using a lot of not properly introduced paleoclimate jargon. Since BG is not a paleo-specific journal and the manuscript topic could also gather interest from outside the paleo-community, it would be valuable if you could make it easier to follow for non paleo-scientists.

- In Sect. 4.2, vegetation changes are directly associated with warmer/colder and wetter/drier conditions. Since the main conclusion of the manuscript is a potentially larger role for $CO_2$ than previously thought, I find this confusing and would recommend a more careful wording when inferring climatic conditions from the vegetation composition. In particular, it should be clarified that the interpretations are just describing changes "felt" by the plants but not necessarily actual climatic changes (unless they are supported by vegetation-independent proxies).

- The role of moisture availability variations remains largely unconstrained in the manuscript which complicates the attribution of the vegetation variability to temperature and/or $CO_2$ changes. The spatial pattern of SST anomalies in the North Atlantic was likely different between HS1 and YD (e.g., Pedro et al., 2022; Weitzel et al., 2024) which could lead to differing precipitation amounts during HS1 and YD. To what extend can you rule out that the moisture availability conditions during the YD were more favorable for forest development than during HS1/LGM? If you cannot constrain the moisture availability changes better at this point, this should be stated as a limitation.

**Specific comments:**

- l. 60: What is meant by "variable moisture condition"?

- l. 91: Do the 5 - 10°C refer to local, zonal mean, or global mean temperature changes? Please specify.

- l. 114 - 127: It would be worth mentioning that there are also previous studies inferring the importance of $CO_2$ changes for the vegetation evolution using multi-proxy approaches (e.g., Gosling et al., 2022; Koutsodendris et al., 2023) and model-data comparison (e.g., Adam et al., 2021). Of particular interest for this study, Koutsodendris et al. (2023) suggests a major role for $CO_2$ in modulating Mediterranean forest growth in Greece.

- l. 152: What simulations are meant here? As far as I know, most simulations of glacial vegetation include the effect of lower $CO_2$.

- Sect. 2: In addition to describing the current regional climate, it would be informative to briefly describe the present-day regional vegetation composition.

- l. 224: Is there a reason for not using IntCal20 instead of the older Marine13 calibration curve? Do you expect this to make a difference?

- Sect. 3.3: Why is the Villarquemado record not included in the regional compilation for which interesting work on the role of $CO_2$ exists (Wei et al., 2021)?

- l. 269: Using the original chronologies of the records will likely lead to some smoothing of the abrupt (centennial-to-millennial scale) variability when computing the regional averages. Could this affect your results?

- l. 278-280: What kind of response function do you use to fit the GAM? Given that pollen percentages are restricted to the interval 0 to 100, a beta or binomial response function is more suitable than a standard Gaussian response function (e.g., Adam et al., 2021; Wei et al., 2020).

- l. 301: Does 0.5°C correspond to a $1\sigma$ standard error?

- l. 336: Does "dry" here refer to the actual climate conditions or the perceived climate conditions of the vegetation (i.e., a combination of moisture availability and $CO_2$)? As stated above, a more careful wording would improve the clarity of this section.

- l. 378-380: Why would the moisture availability change less abruptly than the temperature? Could there also be a role for $CO_2$ or internal vegetation dynamics in explaining the delayed response of the vegetation?

- l. 441-443: Could the presence of large herbivores also play a role in promoting heathland rather than forests (Zhu et al., 2018)?

- l. 467: Why should the precession maximum specifically trigger heathland rather than forest development?

- l. 502: I don't understand why the bias could extend to vegetation reconstructions as these only associate pollen assemblages with vegetation composition. Are you mixing up reconstructions and simulations here? For simulations, it is of course important to account for $CO_2$ changes (which most of them do).

- l. 568-569: What is meant by "in a general assumption"?

- l. 593-597: My understanding of the methodology of Shao et al. (2018) is that they account for $CO_2$ changes by deriving transition matrices from the simulation output of Woillez et al. (2011). If this is correct, it is expected that the two studies agree on the influence of $CO_2$ on the LGM vegetation.

- l. 602: What do you mean by "with some seasonality"?

- l. 644-647: Under which baseline temperature was this experiment conducted? Is the effect dependent on the background temperature?

- Fig. 3: Equating the pollen zones and the geologic periods YD, BA, and HS1 can be misleading since the start and end dates of the pollen zones do not coincide with the official definitions of the geologic periods. For example, pollen zone 4 ending later than the actual YD is not in agreement with the SST reconstruction reaching Holocene level temperatures already during the later stages of pollen zone 4. Therefore, I would recommend to display the limits of the pollen zones separately from the limits of YD, BA, and HS1 in Fig. 3.

- Fig. 4: Based on what data are the schematics for moisture during LGM, HS1, BA, and YD assigned? Do the schematics represent absolute values, anomalies, or trends?

- Table S1: The Top Age for zone 4 should be 11050.

- Fig. S1 and S2: These two figures are rather blurry. Can you improve their resolution?

**Technical comments:**

- I kindly ask you to check the manuscript for typos and grammatical errors again. In particular, there are a number of missing spaces between words (e.g., "ofthe" in l. 178, "elements(Fig.", in l. 366, and "anda" in l. 366).

- The spelling and use of age units differs throughout the manuscript (e.g., "ka" in l. 9, "cal yr BP" in l. 200, "Ka" in l. 560). I kindly ask you to define an age unit in the introduction and use it throughout the manuscript.

**References**

Adam, M., Weitzel, N., and Rehfeld, K.: Identifying Global-Scale Patterns of Vegetation Change During the Last Deglaciation From Paleoclimate Networks, Paleoceanography and Paleoclimatology, 36, e2021PA004 265, https://doi.org/10.1029/2021PA004265, 2021.

Gosling, W. D., Miller, C. S., Shanahan, T. M., Holden, P. B., Overpeck, J. T., and Van Langevelde, F.: A stronger role for long-term moisture change than for $CO_2$ in determining tropical woody vegetation change, Science, 376, 653–656, https://doi.org/10.1126/science.abg4618, 2022.

Koutsodendris, A., Dakos, V., Fletcher, W. J., Knipping, M., Kotthoff, U., Milner, A. M., Müller, U. C., Kaboth-Bahr, S., Kern, O. A., Kolb, L., Vakhrameeva, P., Wulf, S., Christanis, K., Schmiedl, G., and Pross, J.: Atmospheric CO2 forcing on Mediterranean biomes during the past 500 kyrs, Nature Communications, 14, 1664, https://doi.org/10.1038/s41467-023-37388-x, 2023.

Pedro, J., Andersson, C., Vettoretti, G., Voelker, A., Waelbroeck, C., Dokken, T., Jensen, M., Rasmussen, S., Sessford, E., Jochum, M., and Nisancioglu, K.: Dansgaard-Oeschger and Heinrich event temperature anomalies in the North Atlantic set by sea ice, frontal position and thermocline structure, Quaternary Science Reviews, 289, 107 599, https://doi.org/10.1016/j.quascirev.2022.107599, 2022.

Shao, Y., Anhäuser, A., Ludwig, P., Schlüter, P., and Williams, E.: Statistical reconstruction of global vegetation for the last glacial maximum, Global and Planetary Change, 168, 67–77, https://doi.org/10.1016/j.gloplacha.2018.06.002, 2018.

Wei, D., Prentice, I. C., and Harrison, S. P.: The climatic space of European pollen taxa, Ecology, 101, https://doi.org/10.1002/ecy.3055, 2020.

Wei, D., González-Sampériz, P., Gil-Romera, G., Harrison, S. P., and Prentice, I. C.: Seasonal temperature and moisture changes in interior semi-arid Spain from the last interglacial to the Late Holocene, Quaternary Research, 101, 143–155, https://doi.org/10.1017/qua.2020.108, 2021.

Weitzel, N., Andres, H., Baudouin, J.-P., Kapsch, M.-L., Mikolajewicz, U., Jonkers, L., Bothe, O., Ziegler, E., Kleinen, T., Paul, A., and Rehfeld, K.: Towards spatio-temporal comparison of simulated and reconstructed sea surface temperatures for the last deglaciation, Climate of the Past, 20, 865–890, https://doi.org/10.5194/cp-20-865-2024, 2024.

Woillez, M.-N., Kageyama, M., Krinner, G., de Noblet-Ducoudré, N., Viovy, N., and Mancip, M.: Impact of CO2 and climate on the Last Glacial Maximum vegetation: results from the ORCHIDEE/IPSL models, Climate of the Past, 7, 557–577, https://doi.org/10.5194/cp-7-557-2011, 2011.

Zhu, D., Ciais, P., Chang, J., Krinner, G., Peng, S., Viovy, N., Peñuelas, J., and Zimov, S.: The large mean body size of mammalian herbivores explains the productivity paradox during the Last Glacial Maximum, Nature Ecology & Evolution, 2, 640–649, https://doi.org/10.1038/s41559-018-0481-y, 2018.

---

## Author Comment (AC2)

RESPONSE TO THE ANONYMOUS REVIEWERS

We thank the reviewers for their positive and helpful comments on our manuscript. We proposed to revise the paper as suggested.

**Response to anonymous reviewer 2**

**General comments:**
**…However, the paper is too descriptive, the interpretation of lower-concentration CO2 impacts on vegetation is overly general, and some key information is missing.**

**Response:** We appreciate the reviewer's insightful comments and will conduct a thorough proofread to enhance clarity and reduce excessive descriptiveness, particularly in relation to the data presentation.
Regarding the effects of low atmospheric $CO_2$ on vegetation, we recognize the need to strengthen our discussion. We will build on the points already raised in the manuscript,
   a) we highlight the impact of $CO_2$ limitations on photosynthesis and plant growth (e.g., L134, 161), and we also discuss the reduced water-use efficiency (WUE) associated with lower $CO_2$ levels, explaining how decreased $CO_2$ concentrations affect stomatal regulation, leading to increased water loss and heightened vulnerability to drought stress.
   b)  we acknowledge the role of $CO_2$ in shaping plant community composition. Although, we agree that this aspect could be further developed (e.g., L453), particularly in explaining why heathlands may have dominated LGM landscapes. We plan to expand on this by emphasizing that **lower CO₂ levels, combined with cooler conditions, would have favored the establishment of vegetation types adapted to these constraints. The heathland species are resilient, making them well-suited to glacial environments.**
Finally, if the comment "some key information is missing"  pertains to the need for updated references (Rev#2 highlights that "recent papers about pollen-based climate reconstructions (e.g., Izumi and Bartlein, 2016; Chevalier et al., 2020; Wei et al., 2021; Prentice et al., 2022; and Izumi et al., 2023)" we fully acknowledge this and will incorporate this relevant, recent literature in the revised version to strengthen our discussion.

**Specific comments:**

**The authors described that "The study of increased plant growth and global vegetation greening under higher concentrations of pCO2 (CO2 fertilisation) is very topical within discussions of current global climate change (e.g., Piao et**

al., 2019), whilst the inverse scenario (low pCO2) has received less attention."
at line 116-119. However, recent papers about pollen-based climate
reconstructions (e.g., Izumi and Bartlein, 2016; Chevalier et al., 2020; Wei et al.,
2021; Prentice et al., 2022; and Izumi et al., 2023) have discussed the impacts of
lower atmospheric CO2 on vegetation. The reference papers listed here are
pollen-based climate reconstructions using an inverse-modeling approach
related to the papers the authors already cited, Guiot et al. (2000), Wu et al.
(2007), and Prentice et al. (2017). Pollen-based climate reconstructions using
inverse modeling methods have not differed significantly from temperature
reconstructions using conventional statistical methods such as regression
analysis and modern analogue techniques. On the other hand, large differences
are produced in reconstructing hydrological climate from the traditional
methods that tend to overestimate dryness during glacial periods. This may
influence the authors' interpretation of climate from vegetation. As a result, the
authors potentially need to largely rewrite the "Abstract", "Introduction",
"Discussion", and "Conclusion". Moreover, the paper, Piao et al. (2019) is not in
the reference.

**Response:** We greatly appreciate the reviewer's comment, as it highlights an
important and evolving area of research. The development of improved new
methodologies and the continuous refinement of climate reconstructions from pollen
records, particularly under low $CO_2$ conditions, are indeed crucial for improving our
understanding of past environments.

To address the reviewer's concerns, we will integrate the suggested references into
our manuscript, ensuring a more comprehensive discussion of the various approaches
used in pollen-based climate reconstructions. We recognize that the primary issue
related to low $CO_2$ is its influence on moisture availability rather than temperature.
Indeed, this is in agreement with what we have already stated, in different parts of the
manuscript:

*"The Mediterranean region, with its characteristic **annual hydrological deficit and
seasonal water stress**, is a key place for exploring the potential role of pCO_2 limitation
on vegetation growth. Therefore, past vegetation dynamics in this region may be
considered as a significant (inverse) analogue to understand the current impact of
increasing temperature and pCO_2 within semi-arid and arid ecosystems."*

We recognise a need to further our discussion to reflect how different reconstruction
techniques, particularly inverse modeling methods, have contributed to understanding
past climate conditions. To give some context about the exploration done regarding
WA-PLS (noting this exercise was done prior to the published article of Wei et al.,
2021, pers. comm.), we acknowledge the limitations of this approach, and we have
previously tested it on approximately ten Iberian marine and coastal records spanning
the last 23 ka. Our findings indicated two key challenges:
a) The technique systematically pushes reconstructions away from extreme values, a
known issue that we were able to confirm through our tests.

b) Systematic biases were observed in coastal regions, affecting the reliability of the reconstructions.

For those cores, whilst trend comparisons were possible, the presence of modern biases complicated direct interpretation, and the lack of a consistent systematic bias made it difficult to develop an approach to correct the data. In the study of those cores, the use of WA-PLS as a method was not the primary focus of our research, therefore that particular work did not progress further. However, we acknowledge that addressing these methodological challenges here in this current paper is essential for improving the robustness of marine pollen-based reconstructions. The articles of Wei et al., 2021, Prentice et al, 2022 and Cruz-Silva 2023 are able to provide improvements and a better understanding about the $pCO_2$ role.

In summary, we acknowledge the challenges associated with pollen-based climate reconstructions, particularly in the context of low $CO_2$, and we will update our manuscript to integrate recent references and methodological discussions. While our findings align with broader trends reported in the literature, we recognize the need for continued refinement of reconstruction techniques and will ensure that our study appropriately contextualizes these methodological considerations. At the same time we recognize the importance and high value of qualitative datasets, and its information.

**Comment L49-54. The purpose of the authors' study should be to investigate vegetation changes and the effects of pCO2 changes on vegetation on the Iberian Peninsula margin, not to track and compare them with global pCO2 changes.**

**Response:** We appreciate the valuable comment from Reviewer 2. Our intention was to investigate vegetation changes on the Iberian Peninsula margin and assess their relationship with global $pCO_2$ variations. In doing so, we utilized multiple proxies and compared them with global $pCO_2$ records to evaluate how vegetation dynamics responded to these changes—hence our use of the term "tracking." However, we acknowledge that our wording may have led to a different interpretation. To enhance clarity and readability, we will revise the sentence accordingly.

"The high-resolution analysis of terrestrial (pollen, $C_{29}:C_{31}$ organic biomarker) and marine (alkenone-derived Sea Surface Temperature, $C_{37:4}$%, and long-chain n-alkane ratios) indicators, using a direct land-sea comparison at the Iberian margin site Integrated Ocean Drilling Program (IODP) U1385 ("Shackleton site") throughout the Last Glacial Maximum (LGM) and last deglaciation, allowed us to investigate .the Iberian Peninsula vegetation response to major global $pCO_2$ changes/fluctuations."

**Comment L130-132. About stomatal conductance and stomatal density under low pCO2 concentration: Is this correct? Does stomatal density change with application to climate over short periods? Is this the author's idea? If not, please put the paper cited.**

**Response:** We appreciate this valuable comment from Reviewer 2, in fact it misses a reference to back it up. And yes, it is correct there are in fact several studies showing that changes $pCO_2$ can play a strong role by affecting the initiation of stomata, which in turn could impact stomatal density. In this study they present an interesting experiment with elevated and sub-ambient $CO_2$, with durations between 14 days to five years. in which it is possible to detect differences in the number of stomata. I will add for completeness the reference Royer, D. L. (2001). [Stomatal density and stomatal index as indicators of paleoatmospheric CO2 concentration. *Review of Palaeobotany and Palynology*, *114*(1-2), 1-28.] to the sentence highlighted.

**L152-154. I disagree with this sentence, at least about climate reconstructions. In section 3.3, the authors need to describe ecological groups other than the temperate and Mediterranean forests in Figure 3. Which pollen taxa belong to the "semi-desertic taxa" and "heathland", respectively?**

**Response:** We appreciate the valuable comment from Reviewer 2, considering the references given we will update this sentence regarding the contributions highlight in the suggested papers. We will include this information in the Figure 3 caption.

Semi desertic-plants includes Amaranthaceae (Chenopodiaceae), Artemisia and Ephedra.
Heathland includes all the Ericaceae (which includes all the types of Erica sps.) and Calluna sp.

To be more cautious and avoiding overgeneralization we can rephrase it taking out the word "majority". As a note regarding the disagreement, we can just mention that Prentice et al. (2022) as well as Cruz-Silva et al., (2023) are reiterating in their papers there still exist some models which do not contemplate the effect of $CO_2$ in their reconstructions.

**Why do the authors apply a Generalised Additive Model (GAM) to TMF alone? Why not use it for the other ecological groups?**
**Response:** We appreciate this insightful comment.
The decision to apply a Generalized Additive Model (GAM) exclusively to the temperate Mediterranean forest (TMF) group GAM to help reduce noise (counting, chronological) when integrating pollen data from multiple sites into a single curve. We only performed that exercise for the TMF. For the other groups it doesn´t seem necessary.

**L309-316. Is the high production of C29 n-alkanes by trees and shrubs interpreted as a high distribution of these vegetations? The explanations in this text (L309-312) do not help for interpretation for the result.**

**Response:** We thank the reviewer 2 for this question and comment. The understanding of C29 alkanes is that they reflect the production of leaf waxes, generally to protect plants from harsh conditions, hence not necessarily representing the distribution of the species/groups of plants.

We will rephrase this to: "Index values >1are typically considered to reflect higher quantities of $C_{29}$ *n*-alkanes **produced** by trees and shrubs, while values of the index <1 are generally considered to indicate the production of higher quantities of $C_{31}$ *n*-alkanes by grasses and herbaceous plants (Cranwell, 1973; Ortiz et al., 2010; Rodrigues et al., 2009)."

Overall, we will check and revise for clarity the description of the n-alkane results.

**The content in L313-316 should be put before the description of index < 1 or index > 1. L415.**

**Response:** We agree with the reviewer comment; we will implement this change.

**What are the +/- and up/down arrow values for each period (the YD, BA, HS1, and LGM) in the figure (Fig. 4) compared to? Wouldn't it be better to have a quantitative discussion?**

**Response**: The arrows or values are compared to the preceding period, in relation to the referred parameter (we will add this information to the caption of the fig.). A quantitative discussion is not possible because the temperature (except for the SST) and moisture are based on % pollen-based ecological groups. We can instead talk about relative change. Fig. 3 could give a perception of the % values for each parameter.

**L423-426. The authors need to elaborate more, especially on how to read the S.M. Fig.2. Moreover, the figure is difficult to read because of unclear, and its caption is inadequate.**
This approach seems to work if the relationship between vegetation and climate is constant regardless of CO2 concentration. How would we use this figure if the relationship between vegetation and climate changes with changes in CO2 concentration?

**Response:** We sincerely appreciate the reviewer's insightful comments regarding the clarity of **Supplementary Figure 2 (SM Fig.2)**. We acknowledge that the figure's readability could be improved, and we will enhance the visual representation by clearly indicating the parameters analyzed and reinforcing the color contrast to ensure better

comprehension. Also, we will revise the figure caption to provide a clearer explanation of these insights and ensure that readers can more easily interpret the relationships presented.

**In Section 5.1, the authors need to discuss the influences of lower concentration CO2 on vegetation changes, including the more recent papers, especially pollen-based climate reconstruction during the LGM and last glaciation periods.**

**Response:** We appreciate the reviewer's suggestion and acknowledge the importance of incorporating recent studies on the influence of low $CO_2$ concentrations on vegetation changes, particularly in pollen-based climate reconstructions of the LGM and last glaciation periods.

As noted in our previous responses, we will integrate the suggested references into our discussion to provide a more up-to-date perspective on the role of $CO_2$ in shaping past vegetation dynamics. While our current discussion (L493) already considers the impact of moisture and temperature changes, the additional references will strengthen our argument, particularly in highlighting how LGM conditions were likely wetter than previously inferred based on pollen records alone. Given that these recent studies reinforce the idea that pollen-based climate reconstructions often overestimate aridity due to the omission of physiological constraints imposed by low $CO_2$ on plant water-use efficiency (Wei et al., 2021; Prentice et al., 2022; Cruz-Silva et al., 2023), we recognize the value of including them for a more comprehensive discussion.

**In Section 6, what are the key messages of this study? The authors could have described them more briefly and effectively. This is also true in the Abstract.**

**Response:** We appreciate your comment; we will address it by summarizing and including the key messages in bullets points. The key messages for this study will be updated in the manuscript.

- **Vegetation Dynamics Across Key Climate Transitions:** HS1, B-A and YD (to be synthetize)

- **Influence of Low $pCO_2$ on vegetation changes during the LGM:** relatively cold conditions, low seasonality, and wetter but with exacerbated drought stress under low $pCO_2$ further restricted tree growth while promoting heathland expansion, likely due to the moisture-adapted traits of Mediterranean Ericaceae.

- **Critical $pCO_2$ Threshold for Forest Expansion (~225 ppmv):** A) Our study proposes that $pCO_2$ values of ~225 ppmv acted as a critical threshold for forest expansion in the Iberian Peninsula during the last deglaciation. This value is

quite similar with peak in the Mediterranean forest pollen percentages, during MIS13 a decrease in $CO_2$ and a value of 216 ppmv in Oliveira et al. (2020), despite the different insolation conditions. Future modeling efforts should explore the amplitude and thresholds of $pCO_2$ impacts on regional vegetation, including during past cold periods.

Oliveira, D., Desprat, S., Yin, Q., Rodrigues, T., Naughton, F., Trigo, R. M., ... & Goñi, M. F. S. (2020). Combination of insolation and ice-sheet forcing drive enhanced humidity in northern subtropical regions during MIS 13. *Quaternary Science Reviews*, *247*, 106573.

- **Relevance for Present and Future Climate Change:** the findings provide a critical baseline for understanding how arid and semi-arid ecosystems might evolve under rapidly changing $pCO_2$ levels in the present and future.

**The other comments**

**L57. "(HS1)the" to "(HS1), the"**

**Response:** We have adjusted the space as suggested.

**L60. "condition" to "conditions"**

**Response:** We have added an "s" as suggested.

**L64. "mosaic," to "mosaic;"**

**Response:** We have added an ";" as suggested.

**L77. The authors should define the period of "the last deglaciation" here (not L90). Moreover, the authors' definition, from 21 to 6 ka, is not true. The last deglaciation does not include the mid Holocene period.**

**Response:** We thank you the reviewer for this comment. There was a mistake while considering all the period presented in the paper with the deglaciation itself. The begging of the deglaciation occurs at 20-19 ka (e.g. Denton et al 1981; Toucanne et al., 2008; Denton, 2010) spanning to the final episode of Laurentide ice sheet at 7/6.8

ka (e.g. Dyke, 1987; Carlson et al., 2008). Other authors have considered climate during the last deglaciation (20 to 6 cal ka BP)"
https://doi.org/10.5194/cp-6-245-2010, 2010

**L104. "in Northern Hemisphere" to "in the Northern Hemisphere"**

**Response:** We have added "the" as suggested.

**L184. "at centennial-scale resolution" to "at the centennial-scale resolution"**

**Response:** Will change it to "at a centennial-scale resolution" in the middle way that sounds better.

**L202. "the the" to "the"**

**Response:** We have deleted the extra "the" as suggested.

**L205-207. "Köppen classification Csa, with warm summers (around 22°C as the average temperature of the warmest month) mean annual temperatures between 12.5°C and 17.5°C and mean annual precipitation from 400 to 1000 mm/yr."**

**Response:** Not clear about the comment. I will address it if there is more detail.

**L235 and L236. What are "HCl" and "HF"**

**Response:** We understand that some people should not be so familiar with chemistry, including the name of certain acids, as such we have wrote the name of the acids for a better understanding. HCl stands for hydrochloric acid and HF stands for hydrofluoric acid (which is the solution of hydrogen fluoride in water, being a liquid at room temperature).

**L251-252. It is an unclear sentence to me.**

**Response:** This is a normal procedure/calculation, but the explanation may not be clear. When calculating the main sum, we use most of the specimens excluding some taxa (which for different reasons- high production, transport, local and /or aquatic taxa) appeared overrepresented.

Pollen percentages were calculated on the number of pollen grains from terrestrial plants excluding some taxa because of their natural over-representation (Pollen X1 /Main sum). **The percentages of over-represented taxa were calculated on the basis of the main sum plus the counts for that particular individual taxa Pinus / (Main sum + Pinus) and Cedrus / (main sum + Cedrus).** If it helps the understanding I could add the equations for a better understanding.

**L258. Remove "(CONISS)"**

**Response:** We will remove it.
**L258. "(U 1385-1 to 5)" to "(U 1385-1 to 5 in Fig. 3 and Table S1)"**
**Response:** We will edit it.

**L265. "eight pollen records" to "eight marine cores" (?)**
**Response:** We will re-write to "eight marine records"

**L311. "value" to "values"**
**Response:** We will add the "s".

**L392. "inin" to "in"**
**Response:** We will delete the "in"

**L636-643. Basic information on Poaceae and Cyperaceae (Fig. 3g) should be described in Section 3.3 first. It is unclear in the result section why the authors treated Poaceae and Cyperaceae with Fig.3.**

**Response:** I will include this information on section 3. Basically, the Poaceae + Cyperaceae group was because they could be potentially representative of $C_4$ plants (generally better adapted to low $CO_2$ environments), and the curve was to assess if there were any trends compatible with changes with $CO_2$ and $C_{29}/C_{31}$ ratios, being $C_{31}$ related with grasses. In a recent article Casas-Gallego et al. (2025) refers that the actual percentage of $C_4$ is relatively low and contains just ~2.4-5.6% of the vascular plants in SW of Iberia.

Casas-Gallego, M., Postigo-Mijarra, J. M., Sánchez-de Dios, R., Barrón, E., Bruch, A. A., Hahn, K., & Sainz-Ollero, H. (2025). Changes in distribution of the Iberian vegetation since the Last Glacial Maximum: A model-based approach. *Quaternary Science Reviews*, *351*, 109162.

**In Table S1, "1105" to "11050" about the U1385-4 period**
**Response:** We will correct it.

---

## Author Response (AR1)

**Author's response to Editor**

Dear Sandra,

Thank you for your thorough responses to reviewers' comments. Based on those, I have decided your manuscript needs a major revision. Please revise your text accordingly based on your responses to the specific comments. Please also focus on the general clarity of the text mentioned by both reviewers.

Dear editor Petr Kuneš,

Thank you for your comments and for the opportunity to revise our manuscript. We greatly appreciate both your feedback and that of the reviewers, which have been invaluable in significantly improving the quality of our work.

We are pleased to submit the revised version of the manuscript, along with a "Track Changes" document that highlights all the modifications made. Below, we provide detailed responses to each reviewer comment, with the original comments presented in bold, followed by our point-by-point replies. Please note that some of our responses have been slightly revised as well, reflecting adjustments made during manuscript editing to improve clarity and to be more direct.

As requested, we have carefully revised the manuscript throughout to improve its readability and clarity. In particular, the **Abstract** and **Introduction** have undergone substantial rewriting to better communicate the research aims and findings. Additionally, to streamline the presentation, we have merged Section 4.2 and incorporated with previous discussion in Section 5.1, now it stands as results and discussion section 4. This restructuring helps deliver a more cohesive, less repetitive and concise narrative.

We have also introduced the majority of the suggested references, along with several new and relevant citations to strengthen the contextual framework of our study. Furthermore, we have upgraded two figures to enhance their readability and ensure clearer visualization of our results.

We believe these revisions have significantly strengthened the manuscript and addressed all concerns raised by the reviewers. We sincerely thank you and the reviewers again for your constructive feedback, which has contributed greatly to improving the overall quality and impact of our work.

We look forward to your further evaluation.

Kind regards,

Sandra D. Gomes, on behalf of the co-authors

**Author's responses to Reviewer#1:**

**a) General comments:**

While the topic of the manuscript is suitable for publication in BG, the paper is rather difficult to read without a strong paleoclimate background due to using a lot of not properly introduced paleoclimate jargon. Since BG is not a paleo-specific journal and the manuscript topic could also gather interest from outside the paleo-community, it would be valuable if you could make it easier to follow for non-paleo scientists.

**Response:** We sincerely appreciate your feedback and recognize the importance of making this study accessible to a broader interdisciplinary audience. Our research integrates multiple fields, including ecology, palynology, biogeochemistry, quantitative analyses, paleoclimatology, modeling, and functional ecology, all of which contribute to understanding the complex interactions between climate change and various environmental variables over time.

We acknowledge that some paleoclimate-specific terminology may not be familiar to all readers, particularly those outside the paleo-community. Given that *Biogeosciences* is not exclusively a paleo-focused journal, **we have defined key** paleoclimate terms and concepts more clearly to ensure accessibility for a wider readership.

Thank you for your constructive comments. We have improved the manuscript to better serve the diverse scientific community interested in land-ocean ecosystem interactions and their broader implications within the scope of *Biogeosciences*.

In Sect. 4.2, vegetation changes are directly associated with warmer/colder and wetter/drier conditions. Since the main conclusion of the manuscript is a potentially larger role for CO2 than previously thought, I find this confusing and would recommend a more careful wording when inferring climatic conditions from the vegetation composition. In particular, it should be clarified that the interpretations are just describing changes" felt" by the plants but not necessarily actual climatic changes (unless they are supported by vegetation-independent proxies).

**Response:** The paleoclimatic conditions inferred in our study are based on pollen-derived vegetation groups, which serve as indicators of variations in temperature, precipitation, and other climate-related factors. Since atmospheric CO2 strongly influences moisture availability, our interpretations provide a qualitative assessment of relative climate changes rather than absolute quantitative reconstructions. To improve clarity and avoid repetition, we have merged this section with the previous Section 5.2, integrating the results and discussion within the new Section 4.1. Additionally, we have revised the Methods section (Lines 263–266) to explicitly clarify the scope of our climate inferences as follows: "These groups reflect present-day vegetation-climate relationships, allowing inferences about dry, cold, warm, or moist conditions. As such, our pollen data reflect ecological responses rather than absolute quantitative climate parameters (Williams et al., 2001)."

Williams, J.W., Shuman, B.N., Webb, T., Bartlein, P.J., and Leduc, P.L.: Late-Quaternary vegetation dynamics in North America: scaling from taxa to biomes, *Ecological*

Title: Rising atmospheric CO2 concentrations: the overlooked factor promoting SW Iberian Forest development across the LGM and the last deglaciation?

Ms EGUSPHERE-2024-3334 | Research article

*Monographs*, 71, 305–331, https://doi.org/10.1890/0012-9615(2001)071[0305:LQVDIN]2.0.CO;2, 2001.

The role of moisture availability variations remains largely unconstrained in the manuscript which complicates the attribution of the vegetation variability to temperature and/or CO2 changes. The spatial pattern of SST anomalies in the North Atlantic was likely different between HS1 and YD (e.g., Pedro et al., 2022; Weitzel et al., 2024) which could lead to differing precipitation amounts during HS1 and YD. To what extend can you rule out that the moisture availability conditions during the YD were more favorable for forest development than during HS1/LGM? If you cannot constrain the moisture availability changes better at this point, this should be stated as a limitation.

**Response:** Moisture availability, alongside temperature, is inferred from our pollen-based vegetation groups. Pollen studies are essential in this context, as they enable reconstruction of past environmental conditions by leveraging the known ecological requirements of modern plant communities. As previously explained, the presence of steppe indicators is interpreted as evidence of drier conditions, whereas the transition toward heathland and forest assemblages suggests a shift to wetter environments.

We acknowledge the absence of independent precipitation proxies, as noted in line 544, and we have addressed this limitation in our discussion (lines 552–557) by considering the complementary information provided by climate simulations from Cutmore et al. (2021). We have updated the references accordingly to reflect these additions.

Cutmore, A., Ausín, B., Maslin, M., Eglinton, T., Hodell, D., Muschitiello, F., ... & Tzedakis, P. C.: Abrupt intrinsic and extrinsic responses of southwestern Iberian vegetation to millennial-scale variability over the past 28 ka. J. Quat. Sci., 37(3), 420-440, <a href="https://doi.org/10.1002/jqs.3392">https://doi.org/10.1002/jqs.3392</a>, 2022.

**b) Specific comments:**

I. 60: What is meant by "variable moisture condition"?

Response: We change it to "generally wetter conditions" (now Line 59)

**I. 91: Do the 5- 10°C refer to local, zonal mean, or global mean temperature changes? Please specify.**

**Response:** The global temperature increases of 5 to 10 °C, depending on latitude, reported by Bard et al. (1987), Alley and Clark (1999), and Clark et al. (2012), are based primarily on sea surface temperature (SST) reconstructions derived from alkenone proxies, generally calibrated using the Müller et al. (1998) approach. These reconstructions represent *global mean SST*. To improve precision in our terminology, we have clarified this in Line 75 by specifying "global mean temperature representing global mean SST."

l. 114- 127: It would be worth mentioning that there are also previous studies inferring the importance of  $CO_2$  changes for the vegetation evolution using multi-proxy approaches (e.g., Gosling et al., 2022; Koutsodendris et al., 2023) and model-data comparison (e.g., Adam et al., 2021). Of particular interest for this study, Koutsodendris et al. (2023) suggests a major role for  $CO_2$  in modulating Mediterranean forest growth in Greece.

**Response:** We have introduced relevant information from *Gosling et al.* (2022) at Line 419, and from *Koutsodendris et al.* (2023) at Lines 125, 700 and 704 particularly within the Introduction and Discussion sections. Additionally, we have incorporated further references addressing multi-proxy approaches to provide broader context and strengthen our interpretations, and we have updated the reference list accordingly.

- Clément, C., Martinez, P., Yin, Q., Clemens, S. C., Thirumalai, K., Prasad, S., Anupama, K., Su, Q., Lyu, A., Grémare, A., & Desprat, S.: Greening of India and revival of the South Asian summer monsoon in a warmer world. Commun. Earth Environ., 5(1), 685, 2024.
- Gosling, William D., Charlotte S. Miller, Timothy M. Shanahan, Philip B. Holden, Jonathan T. Overpeck, and Frank van Langevelde: A Stronger Role for Long-Term Moisture Change Than for CO2 in Determining Tropical Woody Vegetation Change. Science 376 (6593): 653–56. https://doi.org/10.1126/science.abg4618, 2022.
- Izumi, K., and Bartlein, P. J.: North American paleoclimate reconstructions for the Last Glacial Maximum using an inverse modeling through iterative forward modeling approach applied to pollen data. Geophys. Res. Lett, 43(20), 10-965, https://doi.org/10.1002/2016GL070152, 2016.
- Koutsodendris, A., Dakos, V., Fletcher, W. J., Knipping, M., Kotthoff, U., Milner, A. M., Müller, U. C., Kaboth-Bahr, S., Kern, O. A., Kolb, L., Vakhrameeva, P., Wulf, S., Christanis, K., Schmiedl, G., and Pross, J.: Atmospheric CO2 forcing on Mediterranean biomes during the past 500 kyrs, *Nat. Commun.* 14, 1664, https://doi.org/10.1038/s41467-023-37388-x, 2023.
- Wei, D., González-Sampériz, P., Gil-Romera, G., Harrison, S. P., and Prentice, I. C.: Seasonal temperature and mois ture changes in interior semi-arid Spain from the last inter glacial to the Late Holocene, Quaternary Res., 101, 143–155, https://doi.org/10.1017/qua.2020.108, 2021.

**I. 152: What simulations are meant here? As far as I know, most simulations of glacial vegetation include the effect of lower $CO_2$ .**

**Response:** Thank you for this valuable observation. You are correct that simulations of glacial vegetation include the effects of reduced atmospheric CO2. The Introduction has undergone major restructuring to improve clarity and coherence, to avoid confusion, in this context this sentence was deleted.

**Section 2: In addition to describing the current regional climate, it would be informative to.**

**Response:** We have added a new paragraph to Section 2, "Materials and Environmental Setting," addressing the biogeography of the region (Lines 191–201). This addition is supported by relevant and up-to-date references to provide a more comprehensive environmental context.

Asensi, A. and Díez-Garretas, B.: Coastal Vegetation, in The Vegetation of the Iberian Peninsula, edited by: Loidi, J., Plant and Vegetation, vol. 13, Springer, Cham, Switzerland, pp. 397–432, https://doi.org/10.1007/978-3-319-54867-8 8, 2017.

Rivas-Martínez, S., Penas, Á., del Río, S., Díaz González, T. E., and Rivas-Sáenz, S.: Bioclimatology of the Iberian Peninsula and the Balearic Islands, in The Vegetation of the Iberian Peninsula, edited by: Loidi, J., Plant and Vegetation, vol. 12, Springer, Cham, Switzerland, pp. 29–80, <a href="https://doi.org/10.1007/978-3-319-54784-8">https://doi.org/10.1007/978-3-319-54784-8</a> 2, 2017.

**I. 224: Is there a reason for not using IntCal20 instead of the older Marine13 calibration curve? Do you expect this to make a difference?**

Response: We have re-run the model using the Marine20 calibration curve, incorporating additional radiocarbon dates. Please see the updated Section 3.1, Table 1, and Figure 2 for details.

Heaton, T.J., Köhler, P., Butzin, M., Bard, E., Reimer, R.W., Austin, W.E., Ramsey, C.B, Grootes, P.M., Hughen, K.A, Kromer, B., Reimer, P.J., Adkins, J., Burke, A., Cook, M.S., Olsen, J., Skinner, L.C.: Marine20—the marine radiocarbon age calibration curve (0–55,000 cal BP), *Radiocarbon*,62(4), 779-820, https://doi.org/10.1017/RDC.2020.68, 2020.

**Section 3.3: Why is the Villarquemado record not included in the regional compilation for which interesting work on the role of CO2 exists (Wei et al., 2021)?**

**Response:** We appreciate the reviewer's suggestion to include this reference. Accordingly, we have incorporated relevant information from it at several points in the manuscript: the Introduction (Lines 125, 142, 151) and the Discussion (Line 429). However, we would like to emphasize that this record originates from a significantly different climatic and ecological setting and therefore does not directly represent the ecosystems present in our study region. For example, heathland vegetation is only incipient at the Vilarquermado site. Additionally, the record has some limitations, including lower resolution and hiatuses affecting certain events during the deglaciation period.

Wei, D., González-Sampériz, P., Gil-Romera, G., Harrison, S. P., and Prentice, I. C.: Seasonal temperature and mois ture changes in interior semi-arid Spain from the last inter glacial to the Late Holocene, Quaternary Res., 101, 143–155, https://doi.org/10.1017/qua.2020.108, 2021.

**I. 269: Using the original chronologies of the records will likely lead to some smoothing of the abrupt (centennial-to-millennial scale) variability when computing the regional averages. Could this affect your results?**

**Response:** We appreciate the reviewer's comment and acknowledge that each record carries its own dating uncertainties, which may cause some misalignment when comparing abrupt climatic events. As noted, this could introduce smoothing in regional averages and potentially affect the depiction of centennial-to-millennial-scale variability.

However, in this study, our primary focus is on the overall trend of TMF throughout the deglaciation rather than the precise timing of individual abrupt events across records.

Therefore, we do not believe this smoothing significantly impacts our findings, as our interpretations are based on broader climatic patterns rather than event-scale variability.

I. 278-280: What kind of response function do you use to fit the GAM? Given that pollen percentages are restricted to the interval 0 to 100, a beta or binomial response function is more suitable than a standard Gaussian response function (e.g., Adam et al., 2021; Wei et al., 2020).

**Response:** We appreciate the reviewer's suggestion and acknowledge the importance of selecting an appropriate response function for GAMs when working with pollen percentage data. After testing different models, we ultimately employed a Gaussian GAM with an identity link. While this approach may not be ideal for highly skewed data or values near 0% or 100%, we tested for these issues and determined that the model provided a reasonable fit. We also considered that using REML for smoothness selection helps balance avoiding overfitting and maintaining interpretability.

Nonetheless, we recognize that a quasi-binomial approach might be more appropriate in some cases. If the reviewer has concerns, we are open to revisiting the analysis or, alternatively, removing the graph since it does not significantly affect the overall interpretation of the results.

We have retained the graph and analysis because, although not fundamental to the main discussion, it contributes relevant information illustrating the observed trends. Importantly, the original motivation for producing this synthesis was to demonstrate a consistent wider regional pattern, which increases the geographical relevance of our argument and supports the representativeness of our record.

**I. 301: Does 0.5°C correspond to a 1σ standard error?**

**Response:** Yes, usually it is, unless otherwise specified in the study (now Line 309).

Prahl, F.G., Muehlhausen, L.A., & Zahnle, D.L. (1988). Further evaluation of long-chain alkenones as indicators of paleoceanographic conditions. *Geochimica et Cosmochimica Acta*, 52(9), 2303–2310.

I. 336: Does "dry" here refer to the actual climate conditions or the perceived climate conditions of the vegetation (i.e., a combination of moisture availability and CO2)? As stated above, a more careful wording would improve the clarity of this section.

**Response:** The term "dry" here refers to the perceived climate conditions as inferred from pollen-based ecological groups, grounded in current ecological knowledge. We acknowledge the importance of clarifying this distinction and will refine the wording accordingly. Importantly, whether or not CO2 effects are considered, the presence of steppe indicators reliably signals relatively dry conditions. It is common practice in pollen interpretation to use such ecological groupings as proxies for moisture availability, recognizing that they reflect integrated environmental factors rather than direct climate measurements.

As mentioned previously, we have revised the Methods section (Lines 263–266) to explicitly clarify the scope of our climate inferences as follows:

"These groups reflect present-day vegetation—climate relationships, allowing inferences about dry, cold, warm, or moist conditions. As such, our pollen data reflect ecological responses rather than absolute quantitative climate parameters (Williams et al., 2001)."

Williams, J.W., Shuman, B.N., Webb, T., Bartlein, P.J., and Leduc, P.L.: Late-Quaternary vegetation dynamics in North America: scaling from taxa to biomes, *Ecological Monographs*, 71, 305–331, https://doi.org/10.1890/0012-9615(2001)071[0305:LQVDIN]2.0.CO;2, 2001.

**I. 378-380: Why would the moisture availability change less abruptly than the temperature? Could there also be a role for $CO_2$ or internal vegetation dynamics in explaining the delayed response of the vegetation?**

**Response:** Regarding the role of  $CO_2$ , we do not expect it to have played a significant role during the Bølling–Allerød period. The discussion of the delayed forest response during the BA was primarily related to climate variability and has been addressed in other studies of the Iberian Peninsula (e.g., Naughton et al., 2016). To improve focus and clarity, we have deleted this results after merging the previous Section 4.2 with Section 5 (now section 4.1), thereby concentrating more directly on the main results and their implications in relation to  $CO_2$  impacts.

**I. 441-443: Could the presence of large herbivores also play a role in promoting heathland rather than forests (Zhu et al., 2018)?**

**Response:** Thank you for this interesting question and for providing the reference to Zhu et al. (2018). I agree that large herbivores can influence vegetation dynamics, particularly by helping to maintain heathland landscapes through grazing and browsing. However, I do not think they played a major role in promoting heathlands over forests during the Last Glacial Maximum (LGM), given the combined effects of low atmospheric CO2 concentrations and moisture deficits at that time. Low CO2 would have severely constrained tree growth by limiting photosynthesis, thereby restricting forest expansion regardless of herbivore activity.

Furthermore, in the other records we analyzed — and considering their own local ecology, climate, and fauna — we consistently find evidence of incipient forest development, suggesting that the main limiting factors were climatic and atmospheric rather than primarily related to herbivore pressure.

As mentioned in my previous response, while there are some relevant archaeological studies addressing herbivore impacts, they focus on periods several thousand years older than the time frame considered in our article. Therefore, I think a detailed discussion of this issue would introduce a new topic and deviate from our main focus. However, this would certainly be an interesting avenue to explore in future work, by integrating qualitative archaeological and palaeoecological data with models or simulations, in this particular region.

I. 467: Why should the precession maximum specifically trigger heathland rather than forest development?

**Response:** Thank you for raising this point. I have clarified this aspect in the revised manuscript (see Lines 386–393). Specifically, I explain that the LGM coincided with a precession maximum, which is known to weaken seasonal contrasts (i.e., reduce summer dryness). This climatic configuration has been consistently associated with heathland development in the Iberian Peninsula (Fletcher and Sánchez-Goñi, 2008; Sánchez-Goñi et al., 2008; Margari et al., 2014) across both glacial and interglacial periods (e.g., Oliveira et al., 2017), including the Middle to Late Holocene (Chabaud et al., 2014; Oliveira et al., 2018; Gomes et al., 2020).

Margari, V., Skinner, L.C., Hodell, D.A., Martrat, B., Toucanne, S., Grimalt, J.O., Gibbard, P.L., Lunkka, J.P. and Tzedakis, P.C.: Land-ocean changes on orbital and millennial time scales and the penultimate glaciation. Geology, 42(3), pp.183-186, https://doi.org/10.1130/G35070.1, 2014.

I. 502: I don't understand why the bias could extend to vegetation reconstructions as these only associate pollen assemblages with vegetation composition. Are you mixing up reconstructions and simulations here? For simulations, it is of course important to account for CO2 changes (which most of them do).

**Response**: I agree that pollen-based vegetation reconstructions are based on associating pollen assemblages with vegetation composition, and thus are not directly subject to CO2 biases in the same way as model simulations. To improve clarity, we will rephrase the paragraph and replace the term "vegetation reconstructions" with "simulations of glacial vegetation" in line 439. This more accurately conveys our intention to discuss the emulation of potential modern vegetation and glacial vegetation under different CO2 scenarios.

**I. 568-569: What is meant by" in a general assumption"?**

**Response:** The wording "in a general assumption" was included as part of a sentence to qualify the statement regarding "sufficient moisture," acknowledging that moisture availability is inherently dependent on specific climatic conditions. However, upon re-reading, I agree that this qualification is unnecessary and potentially confusing. The discussion section, including the part addressing the Bølling-Allerød, has undergone significant revisions, and this statement has been rewritten accordingly to improve clarity.

I. 593-597: My understanding of the methodology of Shao et al. (2018) is that they account for  $CO_2$  changes by deriving transition matrices from the simulation output of Woillez et al. (2011). If this is correct, it is expected that the two studies agree on the influence of  $CO_2$  on the LGM vegetation.

**Response:** Thank you for your insightful comment. You are correct that Shao et al. (2018) derive transition matrices from the simulation output of Woillez et al. (2011), and thus both studies highlight the significant role of increased atmospheric CO2 in promoting forest development during the LGM. They are consistent in emphasizing that rising CO2 levels were a crucial driver of vegetation changes at a global scale during this period.

We have rephrase the relevant section (now section 4.1) in the manuscript to make this consistency more explicit. Additionally, we have clarified this topic in the updated discussion

of the Younger Dryas (Section 4.1.4), which does not contradict the points made regarding  $CO_2$  influence on LGM vegetation (Section 4.1.1).

**I. 602: What do you mean by "with some seasonality"?**

**Response:** Thank you for your comment. I agree that the phrase "temperatures with some seasonality" is somewhat vague in a scientific context. To improve clarity and precision, we have rephrased the sentence in the manuscript (lines 573–576) as follows:

"In summary, the persistence of TMF during the YD, despite cold temperatures and seasonal variation (warmer than HS1), seems to be best explained by the combined interaction between sufficient moisture availability, higher atmospheric temperatures—at least during summer (promoting forest development)—and increasing pCO2 (between ~245 and 265 ppmv) (Fig. 4a)."

**I. 644-647: Under which baseline temperature was this experiment conducted? Is the effect dependent on the background temperature?**

**Response:** I appreciate these comments as they highlight a misinterpretation in the original sentence. To better explain the baseline conditions, I have rewritten the passage in the manuscript (Lines 646–653) as follows:

"In laboratory studies, C3 grasses outperform C4 grasses when temperatures rise by 5 to  $15^{\circ}$ C at a low CO2 concentration of 200 ppm. Research on the quantum yield of photosynthesis identified a "crossover temperature"—the point at which C3 and C4 plants perform equally. This crossover depends on both temperature and CO2 levels. Modeling across 0–45°C and CO2 levels from 150–700 ppm shows that whether C3 or C4 plants are favored is determined by the interaction between these two factors, unfortunately humidty was not considered (Ehleringer et al., 1997; Edwards et al., 2010)."

Fig. 3: Equating the pollen zones and the geologic periods YD, BA, and HS1 can be misleading since the start and end dates of the pollen zones do not coincide with the official definitions of the geologic periods. For example, pollen zone 4 ending later than the actual YD is not in agreement with the SST reconstruction reaching Holocene level temperatures already during the later stages of pollen zone 4. Therefore, I would recommend to display the limits of the pollen zones separately from the limits of YD, BA, and HS1 in Fig. 3

**Response:** We have updated Figure 3 to display the boundaries of the pollen zones separately from those of the geologic periods (YD, BA, and HS1). This adjustment preserves the integrity of both datasets while making their temporal distinctions clearer and avoiding potential misinterpretation.

Fig. 4: Based on what data are the schematics for moisture during LGM, HS1, BA, and YD assigned? Do the schematics represent absolute values, anomalies, or trends?

**Response:** The schematics in Figure 4 represent qualitative changes in each parameter based on the interpretation of multiple proxies, including pollen assemblages, sea surface temperatures (SST), and n-alkanes. For moisture, we specifically consider the presence of heathland, tropical montane forest (TMF), and steppe (STE) vegetation groups.

The plus (+) and minus (–) signs indicate fluctuations in moisture conditions throughout each period. Arrows or values are shown relative to the preceding period for the parameter in question. We will add this clarification to the figure caption for improved transparency.

Because temperature and moisture are inferred from percentage-based pollen ecological groups, quantitative values are not directly available; instead, we focus on relative changes. For readers seeking comparative values, Figure 3 presents percentage curves of these groups across each period. To enhance clarity, we have revised the Figure 4 caption as follows:

"Figure 4 – Schematic representation of the relative changes in climate-inferred parameters (precipitation and temperature) based on pollen-vegetation groups, biomarkers, SST, and the physiological contribution of  $CO_2$  for each period, illustrating potential ecosystem scenarios. Temperature inferences are derived from pollen groups (TMF and STE), SST, and n-alkanes; moisture inferences are based on heathland, TMF, and STE groups."

**Table S1: The Top Age for zone 4 should be 11050.**

**Response**: Thank you for highlighting this. However, the age of 11,050 no longer applies, as the new age model provides updated chronological boundaries for zone 4. We have revised Table S1 accordingly to reflect these new age estimates.

**Fig. S1 and S2: These two figures are rather blurry. Can you improve their resolution?**

**Response:** We apologize for the quality issues with these figures, which were also noted by Reviewer 2. We have improved the resolution of Figures S1 and S2 to enhance their clarity and readability.

**Technical comments:**

I kindly ask you to check the manuscript for typos and grammatical errors again. In particular, there are a number of missing spaces between words (e.g., "ofthe" in I. 178, "elements (Fig.", in I. 366, and "anda" in I. 366).

The spelling and use of age units differs throughout the manuscript (e.g., "ka" in I. 9, "cal yr BP" in I. 200, "Ka" in I. 560). I kindly ask you to define an age unit in the introduction and use it throughout the manuscript.

**Response:** We kindly appreciate the technical comments, and we have correct them for consistency. We appreciate your attention to this point and have corrected the manuscript for consistency including in the use of age units. We have adopted *cal ka BP* where calibrated ages are confirmed, and *ka* where calibration is not certain. In the tables and for specifying

age ranges, we prefer to use *cal yr BP*, as it provides the most precise chronological reference without rounding.

**Response to anonymous reviewer 2**

**General comments:**

...However, the paper is too descriptive, the interpretation of lower-concentration CO2 impacts on vegetation is overly general, and some key information is missing.

**Response:** Thank you for these valuable comments. Following this feedback, we have thoroughly revised the manuscript to improve clarity, especially in the abstract and introduction, and by integrating the previous Chapter 4.2 into Section 5.1 in a more concise manner while reducing overly descriptive elements in the data presentation.

Regarding the interpretation of low atmospheric CO2 impacts on vegetation, we have expanded on several key points:

- a) After Line 99, we further discuss how  $CO_2$  limitations affect photosynthesis and plant growth, including the impacts of reduced water-use efficiency (WUE) under low  $CO_2$  concentrations, which influence stomatal regulation, increase water loss, and heighten drought vulnerability.
- b) After Line 113, we expanded the discussion on the role of CO2 in shaping plant community composition, particularly explaining why heathlands may have dominated LGM landscapes given their resilience to low CO2 and cooler conditions.
- Finally, addressing the comment on missing key information, we have incorporated relevant aspects from recent studies (e.g., Izumi and Bartlein, 2016; Chevalier et al., 2020; Wei et al., 2021; Prentice et al., 2022) into both the Introduction and Discussion to strengthen and update the manuscript.
- Chevalier, M., Davis, B. A. S., Heiri, O., Seppä, H., Chase, B. M., Gajewski, K., Lacourse, T., Telford, R. J., Finsinger, W., Guiot, J., Kühl, N., Maezumi, S. Y., Tipton, J. R., Carter, V. A., Brussel, T., Phelps, L. N., Dawson, A., Zanon, M., Vallé, F., Nolan, C., Mauri, A., de Vernal, A., Izumi, K., Holm ström, L., Marsicek, J., Goring, S., Sommer, P. S., Chaput, M., and Kupriyanov, D.: Pollen-based climate reconstruction tech niques for late Quaternary studies, Earth Sci. Rev., 210, 103384, https://doi.org/10.1016/j.earscirev.2020.103384, 2020.
- Chevalier, M., Chase, B. M., Quick, L. J., Dupont, L. M., and Johnson, T. C.: Temperature change in subtropical southeastern Africa during the past 790,000 yr, *Geology*, 49, 71–75, <a href="https://doi.org/10.1130/G47841.1">https://doi.org/10.1130/G47841.1</a>, 2021.
- Izumi, K., and Bartlein, P. J.: North American paleoclimate reconstructions for the Last Glacial Maximum using an inverse modeling through iterative forward modeling approach applied to pollen data. Geophys. Res. Lett, 43(20), 10-965, https://doi.org/10.1002/2016GL070152, 2016.
- Prentice, I. C., Villegas-Diaz, R., and Harrison, S. P.: Account ing for atmospheric carbon dioxide variations in pollen-based reconstruction of past hydroclimates, Global Planet. Change, 211, 103790, https://doi.org/10.1016/j.gloplacha.2022.103790, 2022.
- Wei, D., González-Sampériz, P., Gil-Romera, G., Harrison, S. P., and Prentice, I. C.: Seasonal temperature and mois ture changes in interior semi-arid Spain from the last inter glacial to the Late Holocene, Quaternary Res., 101, 143–155, https://doi.org/10.1017/qua.2020.108, 2021.

**Specific comments:**

The authors described that "The study of increased plant growth and global vegetation greening under higher concentrations of pCO2 (CO2 fertilisation) is very topical within discussions of current global climate change (e.g., Piao et al., 2019), whilst the inverse scenario (low pCO2) has received less attention." at line 116-119. However, recent papers about pollen-based climate reconstructions (e.g., Izumi and Bartlein, 2016; Chevalier et al., 2020; Wei et al., 2021; Prentice et al., 2022; and Izumi et al., 2023) have discussed the impacts of lower atmospheric CO2 on vegetation. The reference papers listed here are pollen-based climate reconstructions using an inverse-modeling approach related to the papers the authors already cited, Guiot et al. (2000), Wu et al. (2007), and Prentice et al. (2017). Pollen-based climate reconstructions using inverse modeling methods have not differed significantly from temperature reconstructions using conventional statistical methods such as regression analysis and modern analogue techniques. On the other hand, large differences are produced in reconstructing hydrological climate from the traditional methods that tend to overestimate dryness during glacial periods. This may influence the authors' interpretation of climate from vegetation. As a result, the authors potentially need to largely rewrite the "Abstract", "Introduction", "Discussion", and "Conclusion". Moreover, the paper, Piao et al. (2019) is not in the reference.

**Response:** As requested, we have carefully revised the manuscript throughout to improve its readability and clarity. In particular, the Abstract and Introduction have undergone substantial rewriting to better communicate the research aims and findings. Additionally, to streamline the presentation, we have merged Section 4.2 and incorporated its discussion into Section 5.1. This restructuring helps deliver a more cohesive, less repetitive, and concise narrative.

To address the reviewers' concerns, we have integrated the suggested references and included several new ones in the manuscript, ensuring a more comprehensive discussion of the various approaches used in pollen-based climate reconstructions.

Comment L49-54. The purpose of the authors' study should be to investigate vegetation changes and the effects of pCO2 changes on vegetation on the Iberian Peninsula margin, not to track and compare them with global pCO2 changes.

**Response:** We appreciate the reviewer's observation and have rephrased the statement accordingly to clarify the study's focus. The revised sentence now reads (Lines 50–51): "This direct land-sea comparison approach allows us to investigate how vegetation on the Iberian Peninsula margin responded to the major pCO2 changes during the last deglaciation."

Comment L130-132. About stomatal conductance and stomatal density under low pCO2 concentration: Is this correct? Does stomatal density change with application to climate over short periods? Is this the author's idea? If not, please put the paper cited.

**Response:** Indeed, the original sentence lacked a supporting reference, which we now include for completeness. It is correct that changes in atmospheric  $pCO_2$  can significantly influence stomatal development. Several experimental studies have demonstrated that both elevated and sub-ambient  $CO_2$  levels can affect the initiation and density of stomata, even

over relatively short timescales. To support this point, we have added the following reference to the revised manuscript:

Royer, D. L.: Stomatal density and stomatal index as indicators of paleoatmospheric CO2 concentration. *Rev. Palaeobot. Palynol.*, **114**, 1–28. <a href="https://doi.org/10.1016/S0034-6667(00)00074-9">https://doi.org/10.1016/S0034-6667(00)00074-9</a>, 2001.

L152-154. I disagree with this sentence, at least about climate reconstructions. In section 3.3, the authors need to describe ecological groups other than the temperate and Mediterranean forests in Figure 3. Which pollen taxa belong to the "semi-desertic taxa" and "heathland", respectively?

**Response:** To avoid overgeneralization, we have rewritten the Introduction and removed the original sentence regarding the "majority" statement. Regarding the reviewer's disagreement, we note that recent studies (e.g., Prentice et al., 2022; Cruz-Silva et al., 2023) emphasize that some models still do not incorporate the effects of  $CO_2$  in their climate reconstructions.

We appreciate the valuable comment from Reviewer 2. Based on the suggested references, we have updated the manuscript to better reflect the ecological groups highlighted in those studies. In particular, we have introduced a detailed description of these groups in the caption of Figure 3, specifying the pollen taxa they include:

- **Semi-desertic taxa**: *Amaranthaceae* (previously *Chenopodiaceae*), *Artemisia*, and *Ephedra*.
- **Heathland taxa**: members of the *Ericaceae* family (including various *Erica* species) and *Calluna* species.

These changes have been incorporated accordingly in the manuscript, including the updated Figure 3 caption.

Why do the authors apply a Generalised Additive Model (GAM) to TMF alone? Why not use it for the other ecological groups?

**Response:** We have retained the graph and analysis because, although not fundamental to the main discussion, it contributes relevant information illustrating the observed trends. Importantly, the original motivation for producing this synthesis was to demonstrate a consistent, wider vegetation/ecological communities—based regional pattern, which increases the geographical relevance of our argument and supports the representativeness of our record.

Regarding the application of the Generalized Additive Model (GAM), we chose to apply it exclusively to the temperate Mediterranean forest (TMF) group to help reduce noise stemming from site locations, counting and chronological uncertainties when integrating pollen data from multiple sites into a single curve. The TMF group exhibited greater variability and complexity that justified this smoothing approach. In contrast, the other ecological groups showed more stable and consistent trends. Therefore, the decision to limit the GAM to the TMF group reflects a targeted effort to improve data clarity where it was most needed.

L309-316. Is the high production of C29 n-alkanes by trees and shrubs interpreted as a high distribution of these vegetations? The explanations in this text (L309-312) do not help for interpretation for the result.

**Response:** We have rephrased this section in Lines 315–326 and provided two hypotheses for interpretation in the Results and Discussion (Section 4.2, Lines 612–623) as follows:

"Instead, we offer two possible interpretations. First, we tentatively infer that the C29/C31 ratio in this setting may reflect an adaptation of plants to aridity and possibly increased wind strength, which together alter the moisture balance. The n-alkanes of leaf waxes are produced to protect plants against water loss during photosynthesis (Post-Beittenmiller, 1996; Jetter et al., 2006). We expect that arid, cold, and windy conditions are more challenging for woody plants, which have more demanding physiological requirements compared to grasses. Therefore, such harsh environments could exert greater stress on woody plants than on herbaceous taxa. Consequently, the increase in the C29/C31 ratio during HS1 and the Younger Dryas could suggest a climatic adaptation of woody plants (TMF and ERI) by increasing the production of C29 leaf waxes as a protective strategy to survive under these challenging conditions (Fig. 3h). Second, the shifts in chain lengths may primarily reflect compositional changes within woody-dominated vegetation, which includes species with diverse ecological tolerances ranging from semi-desert dwarf shrubs such as Artemisia to mesophyll broad-leaved trees."

The content in L313-316 should be put before the description of index < 1 or index > 1. L415.

**Response:** We agree with the reviewer comment; we have implemented this change.

What are the +/- and up/down arrow values for each period (the YD, BA, HS1, and LGM) in the figure (Fig. 4) compared to? Wouldn't it be better to have a quantitative discussion?

**Response**: The schematics in Figure 4 represent qualitative changes in each parameter based on the interpretation of multiple proxies, including pollen assemblages, sea surface temperatures (SST), and n-alkanes. For moisture, we specifically consider the presence of heathland, tropical montane forest (TMF), and steppe (STE) vegetation groups.

The plus (+) and minus (–) signs indicate fluctuations in moisture conditions throughout each period. Arrows or values are shown relative to the preceding period for the parameter in question. We will add this clarification to the figure caption for improved transparency.

Because temperature and moisture are inferred from percentage-based pollen ecological groups, quantitative values are not directly available; instead, we focus on relative changes. For readers seeking comparative values, Figure 3 presents percentage curves of these groups across each period. To enhance clarity, we have revised the Figure 4 caption as follows:

"Figure 4 – Schematic representation of the relative changes in climate-inferred parameters (precipitation and temperature) based on pollen-vegetation groups, biomarkers, SST, and the physiological contribution of  $CO_2$  for each period, illustrating potential ecosystem scenarios. Temperature inferences are derived from pollen groups (TMF and STE), SST, and n-alkanes; moisture inferences are based on heathland, TMF, and STE groups."

L423-426. The authors need to elaborate more, especially on how to read the S.M. Fig.2. Moreover, the figure is difficult to read because of unclear, and its caption is inadequate.

This approach seems to work if the relationship between vegetation and climate is constant regardless of CO2 concentration. How would we use this figure if the

**relationship between vegetation and climate changes with changes in CO2 concentration?**

**Response:** We have updated Supplementary Material Figure S2 to improve clarity and readability. Additionally, we have revised the figure caption to provide a more detailed explanation on how to interpret the figure, ensuring it effectively guides the reader through the key elements presented. The updated figure and caption are now included in the revised Supplementary Material.

The purpose of Figure S2 is to provide a baseline representation of how vegetation responds under current conditions, serving as a starting point for our hypotheses regarding ecological requirements. We acknowledge that the relationship between vegetation and climate is not constant and can be influenced by changing  $CO_2$  concentrations. To address this, we draw Figure 5, which illustrates the constraints on vegetation dynamics through time in relation to temperate and Mediterranean forest (TMF), incorporating the effects of varying temperature and  $CO_2$  levels.

In Section 5.1, the authors need to discuss the influences of lower concentration CO2 on vegetation changes, including the more recent papers, especially pollen-based climate reconstruction during the LGM and last glaciation periods.

**Response:** We have incorporated a series of new references, including the suggested recent studies, to enhance the discussion of how lower CO2 concentrations influenced vegetation changes during the LGM and last glaciation periods. These additions have been integrated into the introduction and Section 4 to provide a more comprehensive and up-to-date perspective on pollen-based climate reconstructions and their implications.

In Section 6, what are the key messages of this study? The authors could have described them more briefly and effectively. This is also true in the Abstract.

**Response:** We have addressed this comment by thoroughly revising multiple sections of the manuscript. Major changes include rewriting the Abstract, Introduction, and Conclusion to clearly and effectively highlight the key messages of the study. Additionally, we merged Section 4.2 with 5.1 to streamline the Results and Discussion, improving clarity and focus on the main findings.

**The other comments**

Due to rewriting several parts, some aspects were removed, while in other cases typos, abbreviations, references, and minor edits were made to improve the text's readability.

L57. "(HS1)the" to "(HS1), the"

L60. "condition" to "conditions"

**Response:** We have added an "s" as suggested.

L64. "mosaic," to "mosaic;"

L77. The authors should define the period of "the last deglaciation" here (not L90). Moreover, the authors' definition, from 21 to 6 ka, is not true. The last deglaciation does not include the mid Holocene period.

**Response:** We thank the reviewer for this important comment. We acknowledge the initial imprecision regarding the definition of the last deglaciation period. The onset of the last deglaciation is generally considered to occur around 20–19 ka (e.g., Denton et al., 1981; Toucanne et al., 2008; Denton, 2010) and extends until the final retreat of the Laurentide Ice Sheet around 7–6.8 ka (e.g., Dyke, 1987; Carlson et al., 2008). While some studies define the last deglaciation broadly as 20 to 6 ka cal BP (e.g., <a href="https://doi.org/10.5194/cp-6-245-2010">https://doi.org/10.5194/cp-6-245-2010</a>, 2010). We have updated the manuscript accordingly to clarify this timeframe and included the relevant references.

L104. "in Northern Hemisphere" to "in the Northern Hemisphere"

L184. "at centennial-scale resolution" to "at the centennial-scale resolution"

Response: L165 changed to "at high (centennial-scale) temporal resolution"

L202. "the the" to "the"

**Response:** We have deleted the extra "the" as suggested.

L205-207. "Köppen classification Csa, with warm summers (around 22°C as the average temperature of the warmest month) mean annual temperatures between 12.5°C and 17.5°C and mean annual precipitation from 400 to 1000 mm/yr."

**Response:** I add a reference to support this information (AEMET, 2011).

Agencia Estatal de Meteorología (AEMET) and Instituto de Meteorologia (IM, Portugal): Atlas Climático Ibérico: Temperatura del aire y precipitación (Experiment Normais 1971–2000), AEMET & IM, Gobierno de España, Madrid and Lisbon, ISBN 978-84-7837-079-5, 2011.

**L235 and L236. What are "HCI" and "HF"**

**Response:** We understand that not all readers may be familiar with the chemical abbreviations used. Therefore, we have replaced the abbreviations with the full names of the acids for clarity. HCl stands for hydrochloric acid, and HF stands for hydrofluoric acid (an aqueous solution of hydrogen fluoride, which is a liquid at room temperature). These changes have been made at Lines 229 and 230.

**L251-252. It is an unclear sentence to me.**

**Response:** We appreciate the comment and agree the original sentence was unclear. The calculation follows a standard procedure, but the explanation could be clearer. When calculating pollen percentages, we use the total count of most terrestrial pollen grains, excluding certain taxa that tend to be overrepresented due to factors like high production, transport, or aquatic origin.

Pollen percentages are calculated as:

- For most taxa: (Pollen count of taxon) / (Main pollen sum) × 100
- For overrepresented taxa (e.g., Pinus, Cedrus), percentages are calculated as: 100 × (Taxon count) / (Main pollen sum + Taxon count)

We included these equations in the manuscript for clarity (Lines 245–248).

L258. Remove "(CONISS)" Response: We removed it.

L258. "(U 1385-1 to 5)" to "(U 1385-1 to 5 in Fig. 3 and Table S1)"

**Response:** We have edited it (now Line 254).

L265. "eight pollen records" to "eight marine cores" (?)

Response: We re-wrote it to "eight marine pollen records" (now Line 274)

L311. "value" to "values" Response: We added the "s".

L392. "inin" to "in"

L636-643. Basic information on Poaceae and Cyperaceae (Fig. 3g) should be described in Section 3.3 first. It is unclear in the result section why the authors treated Poaceae and Cyperaceae with Fig.3. I

**Response:** We have included a brief explanation in Section 3.2 (Lines 257–261) to clarify the treatment of Poaceae and Cyperaceae in Figure 3. It now reads as follows: "In addition to the pollen-based ecological groups, we calculated the sum of Poaceae and Cyperaceae (Fig. 3g), to check the potential importance of C4 plants in the Iberian Peninsula. While most of the present-day Poaceae and Cyperaceae in this region belongs to the C3 plants type (Casas-Gallego et al., 2025), it is possible that C4 plants were more important at other moments in recent Earth history."

Casas-Gallego, M., Postigo-Mijarra, J. M., Sánchez-de Dios, R., Barrón, E., Bruch, A. A., Hahn, K., & Sainz-Ollero, H. (2025). Changes in distribution of the Iberian vegetation since the Last Glacial Maximum: A model-based approach. *Quaternary Science Reviews*, *351*, 109162.

**In Table S1, "1105" to "11050" about the U1385-4 period**

**Response:** We will correct it.

Thank you for highlighting this. However, the age of 11,050 no longer applies, as the new age model provides updated chronological boundaries for zone 4. We have revised Table S1 accordingly to reflect these new age estimates.

---

## Referee Report (RR1)

Gomes et al. "Rising atmospheric CO2 concentrations: the overlooked factor promoting SW lberian Forest development across the LGM and the last deglaciation?"

**General comments**

This study examines pollen and alkenone-based SST records from Iberian Margin Site U1385 to study vegetation changes during the LGM and the deglaciation, linking them to climate and  $CO_2$  variations. A biomarker proxy (the  $C_{29}/C_{31}$  leaf wax ratio) proves effective in reconstructing paleo aridity across the Mediterranean.

Key findings show that low CO2 concentrations (<225 ppm) during the LGM and Heinrich Stadial 1 intensified cold and arid conditions, suppressing forest development while favoring drought-tolerant Ericaceae. Forest expansion during the Bølling-Allerød interstadial was driven by rising CO2 levels and a warmer, wetter climate. Interestingly, despite cooler temperatures during the Younger Dryas, woodland cover persisted—likely sustained by elevated CO2 and increased moisture availability. These results suggest that CO2 exerted its strongest influence on vegetation under cold, low-CO2 conditions, with a critical threshold near 225 ppm. Below this level, forest expansion was severely limited, and climatic stressors were amplified. In contrast, during warmer periods, changes in temperature and precipitation played a more dominant role in shaping vegetation dynamics.

The study emphasizes the role of  $CO_2$  in modulating plant–moisture relationships, offering insights into how future vegetation patterns may respond to anthropogenic  $CO_2$  increases. Further research is also recommended to refine a biomarker that can more effectively distinguish between woody and herbaceous vegetation in Mediterranean ecosystems.

This is a high-quality study that enhances our understanding of vegetation dynamics during the last deglaciation by thoroughly integrating paleo-proxies and highlighting the role of pCO2. The findings are relevant for paleoclimatology, paleoecology, and ecological modeling. However, I believe several revisions are still needed before this manuscript can be published:

- Modify the introduction because it is long and complex, covering multiple concepts (climate dynamics, CO2 physiology, modeling uncertainties, regional paleoecology), which overwhelms readers and hides the main message.
- Strengthen the discussion on why approximately 225 ppmv is a critical threshold (link to plant physiology studies).
- Address biomarker uncertainties with more explicit caveats.
- Typographical and stylistic issues, including mixing British and American English, should be corrected for clarity and consistency (see **line-by-line comments**).

**Specific comments**

The introduction could be improved by addressing several key issues. The logical flow could be better by smoothing abrupt transitions between general and regional topics, using clearer topic sentences and linking phrases. Additionally, some points are repeated unnecessarily, such as discussions of pCO2 effects on water-use efficiency and modeling limitations, which could be made more concise. The research aim, while stated, should be more explicitly and concisely positioned at the end of the introduction to sharpen focus.

The suggested improvements recommend restructuring the content into three or four thematic sections, such as deglaciation and climate background, the role of pCO2 in global and plant physiological dynamics, the regional context of SW Iberia and its vegetation records, and the motivation and goals of the study. Additionally, it advises including a clear statement of objectives at the end, emphasizing the study's purpose to reconstruct high-resolution vegetation changes in SW Iberia using IODP Site U1385 and to evaluate the

influence of  $pCO_2$  on forest dynamics during the Last Glacial Maximum and deglaciation. The feedback also highlights the need to explicitly clarify the study's novelty by identifying gaps not addressed in previous Iberian or global research. Lastly, it suggests reducing repetitive technical jargon, especially regarding the physiological effects of  $pCO_2$ , by summarizing key points concisely instead of reiterating them across multiple paragraphs.

In Methods (3.1. Chronological framework), the authors need to briefly describe why the combination of monospecific and mixed foraminiferal assemblages was used for dating. Are there implications for age reliability? You also clarify whether the chronological uncertainty from the Bacon model was incorporated into subsequent analyses.

In Methods (3.2. Pollen analysis), the reason for excluding aquatic plants and spores from the total could be briefly explained for non-specialist readers.

In Methods (3.3 Compilation of Iberian margin pollen records), the authors can indicate whether chronological alignment or any synchronization across sites was performed, or if all records rely solely on published age models. The GAM model parameters are well-described; however, a brief explanation of why k=30 and sp=0.0001 were chosen would strengthen the statistical justification.

In the Results and Discussion section, although the content is dense and scientifically rich, some parts, especially 4.1.1, 4.1.2, and 4.2, would benefit from clearer and more concise organization. The authors might consider dividing long paragraphs into thematic subsections (e.g., separating observational results from interpretive commentary). There is a slight imbalance between the narrative discussion and data presentation. It could be helpful to include more direct references to quantitative changes (such as percentage increases/decreases,  $\Delta SST$ , pCO $_2$  rise rates) within the text to more explicitly connect interpretations to measured trends.

The identification of a potential  $pCO_2$  threshold (~225 ppmv) for forest development is compelling and well-supported by cross-referenced records. However, some statements treat this threshold as fixed or universal. Consider emphasizing that thresholds may vary by taxa, edaphic conditions, or microclimate, and explicitly acknowledge uncertainties in this area.

The discussion on  $C_{29}/C_{31}$  ratios is thoughtful and cautiously presented, but it could be clearer by organizing it to distinguish between established knowledge, such as the plant physiology of leaf waxes, site-specific observations like correlations in U1385, and interpretive hypotheses, including stress responses versus vegetation shifts. The latter should be explicitly marked as speculative or needing further validation.

The treatment of  $C_3/C_4$  dynamics is detailed and shows skepticism toward simple explanations. The authors correctly highlight the limitations of pollen resolution and suggest promising future directions, such as isotopic or compound-specific research. However, this section could benefit from a summary that outlines the current data limitations and reinforces why  $C_3$  dominance remains the most supported interpretation for SW Iberia during this period.

The hypothesis that pCO2 played a significant and previously underrecognized role in governing vegetation development during the last glacial cycle is compelling and well-supported. This discussion makes a valuable contribution to debates in paleoclimatology, paleoecology, and vegetation modeling, although future data-model comparisons and experimental validations will be crucial to test some of the more speculative physiological mechanisms proposed.

**Line-by-line comments**

L44. "often overlooked" to "often-overlooked"

- L45. "during last deglaciation" to "during the last deglaciation"
- L46. "forest" to "forets"
- L47. "phase, when" to "phase when"
- L53. "of" to "during"
- L57. "suppress" to "suppresses"
- L60. "BP contributed" to "BP, contributed"
- L61. "persistance" to "persistence"
- L99. "in shaping global climate" to "in shaping the global climate"
- L104. "stomata which" to "stomata, which"
- L118. "and typically observed" to "and are typically observed"
- L125. "2023," to "2023;"
- L128. "(e.g. Piao et al., 2020)" to "(e.g. Piao et al., 2020),"
- L147 "resolution, and" to "resolution and"
- L153. "precipitation than" to "precipitation rather than"
- L154. "regional-based" to "region-based"
- L159. "other factors than" to "factors other than"
- L183. "the the" to "the"
- L212. "were" to "was"
- L213. "processed" to "was processed"
- L214. "age-model" to "age model"
- L215. "approach, through" to "approach through"
- L226. "France using" to "France, using"
- L226. "between 3.84 to 1.08" to "between 3.84 and 1.08" or "from 3.84 to 1.08"
- L228. "coarse-sieving" to "coarse sieving"
- L230. "50%) eliminated" to "50%), eliminated"
- L230-231. There appear to be fragmented sentences.
- L260. "plants type" to "plants"
- L261. "Pollen analysis," to "Pollen analysis"
- L293. "k-index obtained" to "k-index was"
- L295. "TMF with" to "TMF, along with"
- L308. "temperatures values" to "temperature values"
- L316. "through the time" to "through time"
- L316. "Eglinton and Hamiltom 1967" no reference list
- L317. "vs grasses plants" to "and grass plants"
- L319. "plants, by increasing" to "plants by increasing"
- L319-320. "increasing leaf wax long chain production" to "increasing the production of long-chain leaf wax"
- L320. ", to reduce" to ", which reduces"

- L325. "dependent" to "depend" or "be dependent"
- L339. "Iberian Peninsula" to "the Iberian Peninsula"
- L352. "Nevertheless" to "Nevertheless."
- L354. "conditions are" to "conditions, are"
- L391. "in addition of" to "in addition to"
- L408. "BIOMEOD" to "BIOME3"
- L428. "Iberia; including" to "Iberia, including"
- L429. "pCO2 yields" to "pCO2, yields"
- L432. "simulations which" to "simulations, which"
- L436. "warmer summer" to "warmer summers"
- L438. "warmer winter temperature" to "warmer winter temperatures"
- L469. "support" to "supports"
- L479. "increase" to "an increase"
- L490. "show" to "shows"
- L500. "are" to "is"
- L516. "was" to "were"
- L531. "Fig 3c" to "Fig. 3c"
- L534. "high latitude" to "high-latitude"
- L545. "a lack of reliability in the speleothem proxies" to "the unreliability of speleothem proxies"
- L553. "nearby" to "near"
- L560. "the forest development" to "forest development"
- L569. "promotor" to "promoter"
- L570. "However" to "However,"
- L591. "thus may" to "may thus"
- L592. "the forest development" to "forest development"
- L597. "using C29/C31 n-alkanes of Site U1385" to "by analysing C29/C31 n-alkanes from Site U1385"
- L603-604. "chain-lengths" to "chain lengths"
- L613. "C29/C31 ratio" to "the C29/C31 ratio"
- L617. "requirments" to requirements
- L618. "Consequently the" to "Consequently, the"
- L630. "different forcing" to "different forcings"
- L630. "the Mediterranean forest development" to "Mediterranean forest development"
- L636-637. "majority C4" to "majority of C4"
- L642. "TMF or STE or  $C_{29}/C_{31}$ " to "TMF, STE, or  $C_{29}/C_{31}$ "
- L649. "Modeling" to "Modelling"
- L650. "favored" to "favoured"

- L651. "humidty" to "humidity"
- L652. "most of the C4" to "most C4"
- L652. "the tropical" to "tropical"
- L662. "although" to ", although"
- L662. "single grain" to "single-grain"
- L675. "ratio) which" to "ratio), which"
- L677. "region)." to "region."
- L679. "palaeonvironment" to "palaeoenvironment"
- L686. "favoring" to "favouring"
- L687. "advantageDuring HS1" to "advantage. During HS1"
- L704. "paleo-data" to "palaeodata"
- L707. " $C_3/C_4$  plants ratio" to "the  $C_3/C_4$  plant ratio"
- L708. "the dynamic" to "the dynamics"

---

## Author Response (AR2)

**Author's response to Editor**

Dear Sandra,

We have now evaluated the manuscript together with two reviewers. Both suggest another moderate revision. I am giving you another chance to incorporate their suggestions and also resolve following issues:

Shorten and focus the introduction (as suggested by rev #2)

Add more detail on constructing the age-depth model (model parameters that are not obvious from methods or results)

Please use the full names of the periods in the subsection titles in the results/discussion section, also use the dating

Add data accessibility statement - where is the primary data, or will it be accessible upon publication?

**Dear editor Petr Kuneš,**

Thank you for your constructive comments and for giving us the opportunity to further improve our manuscript. We are very grateful for the thoughtful feedback provided by both you and the reviewers, which has guided us in strengthening the clarity and focus of the paper.

As requested, we have made the following additional revisions:

- **Introduction**: The section has been shortened and focused to better frame the study within its key objectives and relevance.
- Chronological framework (Methods 3.1): We have added details on the construction of the age-depth model, including model parameters and a clarification of how chronological uncertainties were handled.
- Results/Discussion structure: The subsection titles now use the full names of the climatic periods and their corresponding chronological ranges for greater clarity and consistency.
- **Data accessibility**: We have introduced the following statement:

"The data supporting the findings of this study will be made available upon publication. Interested researchers can access the data by contacting the first author directly or through a publicly accessible data repository."

We are resubmitting the revised manuscript together with a "Track Changes" version that highlights all modifications made. We believe these updates, in combination with our previous revisions, have addressed all the points raised by you and the reviewers.

We sincerely thank you again for your valuable guidance throughout the review process and look forward to your further evaluation.

Kind regards,

Sandra D. Gomes, on behalf of the co-authors

**Author's responses to Reviewer#1:**

I greatly value the time and constructive comments you have provided on the paper I submitted.

The revised manuscript distinguishes more clearly between climatological moisture availability and perceived dryness by plants (due to either changes in climatological moisture availability or changes in water use efficiency in response to different atmospheric CO2 concentrations). However, in my opinion, the term moisture availability should refer only to a climate variable (a function of precipitation, soil moisture, evapotranspiration and/or potential evapotranspiration) and not the perceived dryness by plants. Therefore, I would recommend to not use "moisture availability" when referring to the perceived moisture availability by plants (e.g., in Fig. 4), but either "perceived dryness" or "perceived moisture availability".

**Response:** We thank the reviewer for this valuable comment and agree that the term "moisture availability" could be reserved for climate-related variables (precipitation, soil moisture, evapotranspiration, and/or potential evapotranspiration). After discussion, we considered the suggested term "perceived moisture availability," but we feel that the notion of "perception" could imply sensory mechanisms, which may be misleading in the context of plant physiology. As an alternative, we have adopted the term "plant-available moisture" throughout the manuscript, which we believe better captures the intended meaning while maintaining scientific clarity and avoiding confusion.

- I. 62-66: The last two sentences of the abstract are rather difficult to understand. Maybe you can reformulate them to convey a clearer message.

**Response:** The last two phrases were reformulated to better convey the message " *Our study* suggests that during cold and humid periods (LGM and YD) different pCO2 values lead to contrasting SW Iberian vegetation responses. In contrast, temperature and precipitation changes during periods of relatively high pCO2 play the main role in shaping the distribution and composition of the vegetation."

- I. 75-76: A "global mean temperature increase of 5-10°C" is not in line with recent studies by Annan et al. (2022), Osman et al. (2021), and Tierney et al. (2020). Please update the provided range accordingly.

**Response:** We thank the reviewer for this helpful suggestion. We have updated the statement to align with recent studies. The revised text now reads:

"The last deglaciation, spanning 20–19 cal. kyr BP (e.g., Denton et al., 1981; Toucanne et al., 2008; Denton, 2010) to ~7 cal. kyr BP (e.g., Dyke and Prest, 1987; Carlson et al., 2008), was

marked by a global annual mean surface air temperature increase of ~5°C (Annan et al., 2022), during progressive melting of Northern Hemisphere glaciers. "

This change removes the outdated 5–10°C range and reflects the more constrained estimates provided by recent studies.

- I. 77-79: It should be noted that the warming/cooling during HS1/BA/YD refers mainly to the North Atlantic region (or the Northern Hemisphere). In the Southern Hemisphere, the changes during these periods are (partly) different.

**Response:**

We thank the reviewer for this useful clarification. To address this point, we have specified the geographical context by adding "during progressive melting of Northern Hemisphere glaciers" at the end of the referred statement.

- I. 294: "basis" instead of "basin".

**Response:** We thank the reviewer for noting this typo. The word "basin" has been corrected to "basis."

- I. 363: Consider using "the modern environmental space" instead of "the environmental space"

**Response:** We thank the reviewer for this helpful suggestion. We have revised the text to use *"the modern environmental space"* for clarity, and we have also updated the caption of Fig. S2 accordingly.

- I. 473: I suggest to use "reconstructed" instead of "modelled".

**Response:** We thank the reviewer for this suggestion. The term "modelled" has been replaced with "reconstructed" as recommended.

- I. 1229: In the brackets, it should be "perceived moisture availability" or "perceived dryness" instead of "precipitation"

Response: As discussed previously we changed it to "plant-available moisture".

**Author's responses to Reviewer#2:**

We greatly value the time and constructive comments you have provided on the paper I submitted.

**General comments:**

Modify the introduction because it is long and complex, covering multiple concepts (climate dynamics, CO2 physiology, modelling uncertainties, regional paleoecology), which overwhelms readers and hides the main message.

**Response:** The introduction was shortened and organized to better reflect the paper's message.

Strengthen the discussion on why approximately 225 ppmv is a critical threshold (link to plant physiology studies).

Response: We thank the reviewer for this valuable suggestion. In the revised manuscript, we have strengthened the discussion on why ~225 ppmv represents a critical threshold for vegetation development by integrating insights from plant physiology studies. Specifically, we introduced a new sentence in the *Introduction: "This plasticity in stomatal frequency is considered an adaptive trait that evolved under declining Cenozoic CO2 levels, enabling plants to sustain carbon uptake as concentrations approached glacial minima (~180–190 ppmv), though at the cost of greater water loss (Wagner et al., 1997)." Additionally, in the Discussion* (section 4.1.1), we now state: "However, modelling approaches indicate that in C3 plants, photosynthetic capacity declines sharply once atmospheric CO2 falls below ~300 ppmv, making carbon assimilation increasingly limiting for plant growth (Wagner et al., 1997)." These additions explicitly link the proposed ~225 ppmv threshold to well-established physiological mechanisms, while emphasizing that it should be interpreted as a critical range rather than a fixed universal limit.

**Address biomarker uncertainties with more explicit caveats.**

**Response:** We thank the reviewer for pointing out the need to more explicitly acknowledge biomarker-related uncertainties. In the revised manuscript, we have expanded section 3.4 (Methods) to state:

"Nevertheless, uncertainties remain, since  $U^{\kappa'}_{37}$  SST reconstructions may be affected by calibration biases, seasonal and ecological effects related to coccolithophorid production, and potential lateral transport or diagenetic alteration of alkenones (e.g., Conte et al., 2006; Ausín et al., 2022). Therefore, while the derived SSTs provide reliable insights into large-scale temperature variability, they should be interpreted with caution regarding the magnitude and seasonality of past changes."

This addition highlights the caveats more explicitly while reinforcing the robustness of the trends we report.

Conte, M. H., Sicre, M.-A., Rühlemann, C., Weber, J. C., Schulte, S., Schulz-Bull, D., and Blanz, T.: Global temperature calibration of the alkenone unsaturation index (UK'37) in surface waters and comparison with surface sediments, Geochem. Geophys. Geosyst., 7, Q02005, <a href="https://doi.org/10.1029/2005GC001054">https://doi.org/10.1029/2005GC001054</a>, 2006.

Ausín, B., Haghipour, N., Bruni, E., and Eglinton, T.: The influence of lateral transport on sedimentary alkenone paleoproxy signals, Biogeosciences, 19, 613–627, https://doi.org/10.5194/bg-19-613-2022, 2022.

Typographical and stylistic issues, including mixing British and American English, should be corrected for clarity and consistency (see line-by-line comments).

**Response:** We sincerely appreciate the reviewer's careful attention to typographical and stylistic issues. All spelling, grammar, and punctuation errors have been corrected, and the manuscript has been revised for consistency. We have also standardized the use of English throughout the text to maintain clarity, avoiding mixing British and American conventions.

**Specific comments**

In Methods (3.1. Chronological framework), the authors need to briefly describe why the combination of monospecific and mixed foraminiferal assemblages was used for dating. Are there implications for age reliability? You also clarify whether the chronological uncertainty from the Bacon model was incorporated into subsequent analyses.

**Response:** We have modified the Methods section (3.1 Chronological framework) to clarify the rationale for using a combination of monospecific and mixed foraminiferal assemblages:

"A new set of eleven samples for AMS ¹⁴C analysis was selected primarily from monospecific assemblages of G. bulloides. When sample size requirements could not be met, a mixed assemblage of G. bulloides and G. inflata was used. All samples were processed at the Keck Carbon Cycle AMS Facility, University of California, Irvine (Table 1)."

As standard practice, monospecific *G. bulloides* assemblages are preferred for radiocarbon dating. Mixed assemblages are used only when sample weight is insufficient. While potential age offsets may exist in mixed assemblages, these are generally modest and context-dependent. Previous studies have addressed similar issues, such as Barker et al. (2007) for the North Atlantic and Ausín et al. (2019) for the Iberian margin (though using *G. ruber*). The ERC project *Passenger* also highlights challenges related to age offsets and reservoir differences across deglaciation events.

Regarding the age model, the *Bacon* software incorporates chronological uncertainty by using Bayesian modelling of accumulation rates and the probability distribution of each radiocarbon date. We included calibration using the Marine20 curve, which accounts for reservoir effects.

While the age model does not aim for ultra-high precision in dating well-known deglacial events, it provides a robust and regionally consistent chronological framework suitable for the goals of this study.

In Methods (3.2. Pollen analysis), the reason for excluding aquatic plants and spores from the total could be briefly explained for non-specialist readers.

**Response:** We have revised the Methods section (3.2 Pollen analysis) to provide the requested explanation for non-specialist readers. The sentence now reads:

"Aquatic plants and spores were excluded because their abundant pollen originates in or near water bodies and can be transported far from their source, potentially overrepresenting regional vegetation. Pinus pollen, which is typically overrepresented in marine deposits, is transported by rivers from the Tagus and Sado's watersheds (Naughton et al., 2007). In contrast, the overrepresented *Cedrus* is transported by wind from the Atlas or Rif mountains in Morocco. Both overrepresented taxa were also excluded from the main sum."

In Methods (3.3 Compilation of Iberian margin pollen records), the authors can indicate whether chronological alignment or any synchronization across sites was performed, or if all records rely solely on published age models. The GAM model parameters are well-described; however, a brief explanation of why k = 30 and sp = 0.0001 were chosen would strengthen the statistical justification.

**Response:**

We have revised the Methods section (3.3 Compilation of Iberian margin pollen records) to clarify the chronological framework and GAM justification.

All Iberian margin pollen records rely on published age models; no additional chronological alignment or synchronization across sites was performed. For clarity we add to that particular sentence this "without any additional alignment or synchronization"

The GAM model was fitted using the gam() function of the **mgcv** package (version 1.8.24; Wood, 2017) in R (version 3.6.3; R Core Team, 2020). We used a standard GAM with REML smoothness selection, specifying 30 basis functions (k = 30) and a smoothing parameter of 0.0001 (sp = 0.0001). The relatively high k allowed the model to capture potential nonlinear patterns in the data without overfitting, while the small sp ensured sufficient smoothness; these values were selected after exploratory analysis and diagnostic checks. To assess the validity of the smooth terms and confirm that the basis functions adequately captured the data wiggliness, we applied the gam.check() function. The resulting k-index was greater than 1, and the p-values supported the hypothesis that sufficient basis functions were used. The fitted GAM curves for TMF are presented along with approximate 95% confidence intervals (Simpson, 2018).

In the Results and Discussion section, although the content is dense and scientifically rich, some parts, especially 4.1.1, 4.1.2, and 4.2, would benefit from clearer and more concise organization. The authors might consider dividing long paragraphs into thematic subsections (e.g., separating observational results from interpretive

commentary). There is a slight imbalance between the narrative discussion and data presentation. It could be helpful to include more direct references to quantitative changes (such as percentage increases/decreases,  $\Delta$ SST, pCO2 rise rates) within the text to more explicitly connect interpretations to measured trends.

**Response**: Thank you for this constructive feedback. In response, we have reorganized Sections 4.1.1, 4.1.2, and 4.2 to improve clarity and readability. Long paragraphs have been divided into thematic subsections, starting with vegetation and climate-inferred observations, followed by model-based evidence, and concluding with an interpretive synthesis. This structure allows the reader to follow the progression from data presentation to interpretation more clearly.

All quantitative information, including pollen percentages, SST minima and maxima, and pCO $_2$  changes, was already included in the original manuscript. Our main effort in this revision was therefore focused on simplifying complex sentences, breaking up dense paragraphs, and improving the overall flow of the text. Figure 5 has also been emphasized in the interpretive synthesis to visually integrate TMF, SST, and pCO $_2$  data, helping to guide the reader through the key trends.

We believe these revisions address the reviewer's concerns regarding organization and readability while maintaining the richness of the scientific content.

The identification of a potential pCO2 threshold (~225 ppmv) for forest development is compelling and well-supported by cross-referenced records. However, some statements treat this threshold as fixed or universal. Consider emphasizing that thresholds may vary by taxa, edaphic conditions, or microclimate, and explicitly acknowledge uncertainties in this area.

**Response:** We thank the reviewer for this valuable comment. In the revised manuscript (Section 4.1.2), we have clarified that the  $\sim$ 220–225 ppmv pCO2 values should not be considered a fixed universal threshold, as they likely vary depending on plant taxa, edaphic conditions, and microclimate. We have also emphasized the uncertainties associated with defining strict thresholds, given the lack of experimental studies testing forest development under very low pCO2 levels (most existing work focuses on high pCO2 impacts). To strengthen this point, we now note that modelling approaches indicate photosynthetic capacity in C3 plants declines sharply once atmospheric CO2 concentrations fall below  $\sim$ 300 ppmv, making carbon assimilation increasingly limiting for growth (Wagner et al., 1997). The revised paragraph therefore highlights both the potential role of low pCO2 in constraining forest development during the LGM and the importance of interpreting these values as a critical range rather than a fixed limit.

Wagner F., Below R., de Klerk P., Dilcher D. L., Joosten H., Kürschner W. M. & Visscher H. (1997). A natural experiment on plant acclimation: lifetime stomatal frequency response of an individual tree to annual atmospheric  $CO_2$  increase. Proceedings of the National Academy of Sciences of the United States of America, 93(21), 11705–11708. https://doi.org/10.1073/pnas.93.21.11705

The discussion on C29/C31 ratios is thoughtful and cautiously presented, but it could be clearer by organizing it to distinguish between established knowledge, such as the

plant physiology of leaf waxes, site-specific observations like correlations in U1385, and interpretive hypotheses, including stress responses versus vegetation shifts. The latter should be explicitly marked as speculative or needing further validation.

**Response:** We thank the reviewer for the insightful suggestion. In response, we have reorganized section 4.2 on  $C_{29}/C_{31}$  ratios to clearly distinguish between established knowledge, site-specific observations, and interpretive hypotheses, as suggested. We have also revised the wording regarding the variability of n-alkane production across species and regions. This phrasing emphasizes the limitations of using the  $C_{29}/C_{31}$  ratio as a strict taxonomic proxy while maintaining the context for interpreting site-specific patterns at U1385. The treatment of  $C_3/C_4$  dynamics is detailed and shows scepticism toward simple explanations. The authors correctly highlight the limitations of pollen resolution and suggest promising future directions, such as isotopic or compound-specific research. However, this section could benefit from a summary that outlines the current data limitations and reinforces why  $C_3$  dominance remains the most supported interpretation for SW Iberia during this period.

**Response:** We thank the reviewer for this constructive comment. Following the suggestion, we have added a concluding paragraph to this section that, in our understanding, flows from the current uncertainties and highlights the need for future isotopic and biomarker research, while also emphasizing that the available evidence supports  $C_3$  dominance in SW Iberia during deglaciation. The added text reads:

"In summary, although future isotopic and biomarker approaches hold great promise for resolving  $C_3/C_4$  dynamics, the current evidence strongly supports  $C_3$  dominance in SW Iberia during deglaciation. This interpretation is consistent with the modern distribution of plants in the region, where less than 10% of grasses are  $C_4$  (Casas-Gallego et al., 2025), and with the prevailing cool and humid conditions of the LGM and YD, which favour  $C_3$  over  $C_4$  photosynthesis. Thus, while acknowledging the limitations of pollen-based proxies, the available data indicate that  $C_3$  plants were the dominant contributors to the Iberian vegetation signal."

The hypothesis that pCO2 played a significant and previously underrecognized role in governing vegetation development during the last glacial cycle is compelling and well-supported. This discussion makes a valuable contribution to debates in paleoclimatology, paleoecology, and vegetation modelling, although future data-model comparisons and experimental validations will be crucial to test some of the more speculative physiological mechanisms proposed.

**Response:** We fully agree with this assessment. In our effort to explore this further, we contacted some vegetation modellers to attempt simulations for the YD and LGM. However, as we understood, the temporal resolution and computational resources required made such simulations challenging at that time. We remain very open to collaborating on future model-data comparisons, particularly at a regional scale, to further test the proposed physiological mechanisms.

**Line-by-line comments**

Title: Rising atmospheric CO2 concentrations: the overlooked factor promoting SW lberian Forest development across the LGM and the last deglaciation?

Ms EGUSPHERE-2024-3334 | Research article

**Response:** We sincerely appreciate the technical comments regarding spelling, grammar, and punctuation, and have corrected them for consistency in the revised manuscript.